# Munc18 and Munc13 serve as a functional template to orchestrate neuronal SNARE complex assembly

Shen Wang [1], Yun Li[1], Jihong Gong[1], Sheng Ye[2], Xiaofei Yang[3], Rongguang Zhang[2,4] & Cong Ma[1]

The transition of the Munc18-1/syntaxin-1 complex to the SNARE complex, a key step involved in exocytosis, is regulated by Munc13-1, SNAP-25 and synaptobrevin-2, but the underlying mechanism remains elusive. Here, we identify an interaction between Munc13-1 and the membrane-proximal linker region of synaptobrevin-2, and reveal its essential role in transition and exocytosis. Upon this interaction, Munc13-1 not only recruits synaptobrevin-2-embedded vesicles to the target membrane but also renders the synaptobrevin-2 SNARE motif more accessible to the Munc18-1/syntaxin-1 complex. Afterward, the entry of SNAP-25 leads to a half-zippered SNARE assembly, which eventually dissociates the Munc18-1/syntaxin-1 complex to complete SNARE complex formation. Our data suggest that Munc18-1 and Munc13-1 together serve as a functional template to orchestrate SNARE complex assembly.

[1] Key Laboratory of Molecular Biophysics of the Ministry of Education, College of Life Science and Technology, and the Collaborative Innovation Center for Brain Science, Huazhong University of Science and Technology, 430074 Wuhan, China. [2] National Laboratory of Biomacromolecules, Institute of Biophysics, Chinese Academy of Sciences, 100101 Beijing, China. [3] Key Laboratory of Cognitive Science, Hubei Key Laboratory of Medical Information Analysis and Tumor Diagnosis and Treatment, Laboratory of Membrane Ion Channels and Medicine, College of Biomedical Engineering, South-Central University for Nationalities, 430074 Wuhan, China. [4] National Center for Protein Science Shanghai, Institute of Biochemistry and Cell Biology, Shanghai Institutes for Biological Science, Chinese Academy of Sciences, 201203 Shanghai, China. These authors contributed equally: Shen Wang, Yun Li. Correspondence and requests for materials should be addressed to C.M. (email: cong.ma@hust.edu.cn)

In neurons, neurotransmitter release by synaptic exocytosis is accomplished by tethering of synaptic vesicles at active zones on the plasma membrane; docking/priming of vesicles to a maturation stage that enables them fusion competent; and fast fusion of vesicles with the plasma membrane in response to $Ca^{2+}$ influx[1]. The core fusion machinery for synaptic exocytosis comprises three neuronal soluble *N*-ethylmaleimide-sensitive factor activating protein receptor (SNARE) proteins, syntaxin-1 (Syx1), SNAP-25 (SN25), and synaptobrevin-2 (Syb2). Syx1 and SN25 are localized on the plasma membrane, and Syb2 resides in synaptic vesicles[2,3]. The SNARE proteins are characterized by conserved stretches of 60–70 amino acids termed SNARE motifs[4]. Isolated SNARE motifs are largely unstructured but adopt a folded structure upon formation of a stable four SNARE helical bundle comprising 15 hydrophobic binding layers (layers −7 to −1 and +1 to +8) and an ionic binding layer (layer 0) at the center of the bundle[5,6]. In addition, Syx1 and Syb2 both contain a linker region (LR) connecting their SNARE motifs to their membrane anchors. The pairing of the three SNAREs from the N-terminal SNARE motifs to the C-terminal anchors leads to the formation of the SNARE complex, which brings membranes into close proximity and results in final fusion[7–10]. Although the SNAREs alone can induce liposome fusion in vitro, the fusion rate is rather slow (requiring hours for completion), which is difficult to reconcile with the millisecond time scale that is characteristic of exocytosis[11]. To achieve the observed speed and fidelity of exocytosis, a number of fusion components are required to act in concert with the SNAREs to enable the exquisite regulation of SNARE complex formation[12,13].

Particularly crucial among these components are the Sec1-Munc18 (SM) protein Munc18-1 and the complexes associated with tethering containing helical rods (CATCHR)-related protein Munc13[14–16], which play a central function in exocytosis through orchestrating SNARE complex assembly[12,13]. It is known that Munc18-1 initially locks Syx1 in a closed conformation to inhibit SNARE complex assembly[17,18]. Increasing evidence has revealed that the MUN domain of Munc13-1 catalyzes the transition from the Munc18-1/Syx1 complex to the SNARE complex in the presence of SN25 and Syb2[19–22]. In addition, multiple studies suggest that during the transition from the Munc18-1/Syx1 complex to the SNARE complex, Munc18-1 domain 3 adjusts its conformation from a bent structure compatible with the closed Syx1 to an extended structure compatible with the open Syx1[23–27]. Moreover, consistent with a Vps33/Nyv1 model[28], the extended structure of Munc18-1 domain 3 is favorable for Syb2 binding[29,30], suggesting that Munc18-1 can serve as a template to bring Syx1 and Syb2 into close proximity for proper register[28]. Furthermore, Munc18-1 in coordination with Munc13-1 has been proposed to chaperone proper SNARE assembly[20,31,32], rendering the docked/primed vesicles competent for $Ca^{2+}$-triggered fusion. Hence, all these studies have indicated a Munc18–Munc13 route to SNARE complex assembly[33]. Despite these advances, the sequential and distinct actions of Syb2 and SN25 during this pathway have been elusive, and dissecting these actions is of primary importance for understanding the exquisite regulation of exocytosis.

Here, we discover an interaction between the MUN domain of Munc13-1 and the membrane-proximal LR of Syb2, and reveal its importance in the transition from the Munc18-1/Syx1 complex to the SNARE complex and in vesicle docking/priming. Upon this interaction, Munc13-1 recruits Syb2-embedded vesicles onto the target membrane and renders the SNARE motif of Syb2 more accessible to the Munc18-1/Syx1 complex. With the arrival of SN25, a half-zippered SNARE assembly results in the release of Syx1 from Munc18-1 tight clamping, eventually leading to full SNARE complex formation. These data present a coherent picture of how Munc18-1 and Munc13-1 cooperate with the three SNAREs to orchestrate SNARE complex assembly.

## Results

**A half-zippered SNARE assembly gates the transition.** Aided by the crystal structure of the SNARE complex[6] (Fig. 1a), we designed a series of mutations or truncations on the SNAREs (all mutations or truncations used throughout the study are summarized in Supplementary Table 1), and tested their potential defects in MUN-catalyzed transition from the Munc18-1/Syx1 complex to the SNARE complex. With a native polyacrylamide gel electrophoresis (PAGE) assay described previously[22], MUN-catalyzed transition to the SNARE complex was reflected by the depletion of a characteristic band corresponding to the Munc18-1/Syx1 complex on the gel (Fig. 1b). In this assay, disappearance of the Munc18-1/Syx1 complex band can only be detected when all the components were included (Fig. 1c). Importantly, the immunoblotting data confirmed that the appearance of the SNARE complex was accompanied by the disappearance of the Munc18-1/Syx1 complex (Supplementary Fig. 1, lane 14), despite the coexistence of SNARE assembly intermediates that run slower than the actual SNARE complex as indicated on the native PAGE gel (Fig. 1c, lane 14). The evidence indicates that the formation of the SNARE complex corresponds to the dissociation of the Munc18-1/Syx1 complex. These data also suggest that the transition from the Munc18-1/Syx1 complex to the SNARE complex requires not only Munc13 catalysis but also the presence of SN25 and Syb2. Consistent with previous results[19–22], the Munc13-1 MUN domain efficiently promoted the transition of the Munc18-1/Syx1 (residues 1–261) complex to the SNARE complex in the presence of SN25 (residues 1–206) and Syb2 (residues 29–96), as indicated by the disappearance of the Munc18-1/Syx1 complex on the gel (Fig. 1c). Compared to Syb2, the truncations $Syb2^{\Delta-7}$ and $Syb2^{\Delta-6}$, but not the mutation $Syb2^{-7}$ (L32A) (Supplementary Table 1), severely hindered the transition, suggesting that layers −7 and −6 of Syb2 are required for the transition (Fig. 1d, e), possibly to drive SNARE N-terminal nucleation and zippering. To test this notion, we turned to a Syb2-peptide (residues 49–96, $Syb2^{49–96}$) displacement assay[8], in which the ability of Syb2 and its truncations/mutations to drive SNARE N-terminal nucleation and zippering can be characterized by assessing the disassembly of $Syb2^{49–96}$ from the ΔN-complex $(Syx1/SN25/Syb2^{49–96})$ (interpreted in Supplementary Fig. 2). Similar to the transition results, compared to Syb2, the truncations $Syb2^{\Delta-7}$ and $Syb2^{\Delta-6}$, but not the mutation $Syb2^{-7}$ (L32A), exhibited impaired ability for the displacement of $Syb2^{49–96}$ from the pre-assembled ΔN-complex (Supplementary Fig. 2a). These data indicate an inherent need for the −7 and −6 layers of Syb2 to drive SNARE N-terminal nucleation and zippering, implying their essential role in the transition. As a consequence, Munc18–Munc13 route to SNARE complex assembly would be expected to initiate with N-terminal nucleation and proceeds in an N- to C-terminal zippering manner.

Consistent with this notion, it was recently reported that the MUN domain does not dissociate the closed Munc18-1/Syx1 complex, but rather induces a conformational change in the Syx1 LR and the adjacent N-terminal end of the Syx1 SNARE motif, which provides an ideal template for SN25 and Syb2 nucleation at the N-terminal end[21]. In fact, propagation of the four-helix SNARE bundle towards the C-terminal end is believed to provide energy to dissociate the Munc18-1/Syx1 complex[21]. Hence, we sought to determine at which layer N- to C-zippering of the SNAREs must propagate to enable the transition from the Munc18-1/Syx1 complex to the SNARE complex. Considering that both Syx1 and Syb2 are implicated in Munc18-1 interaction

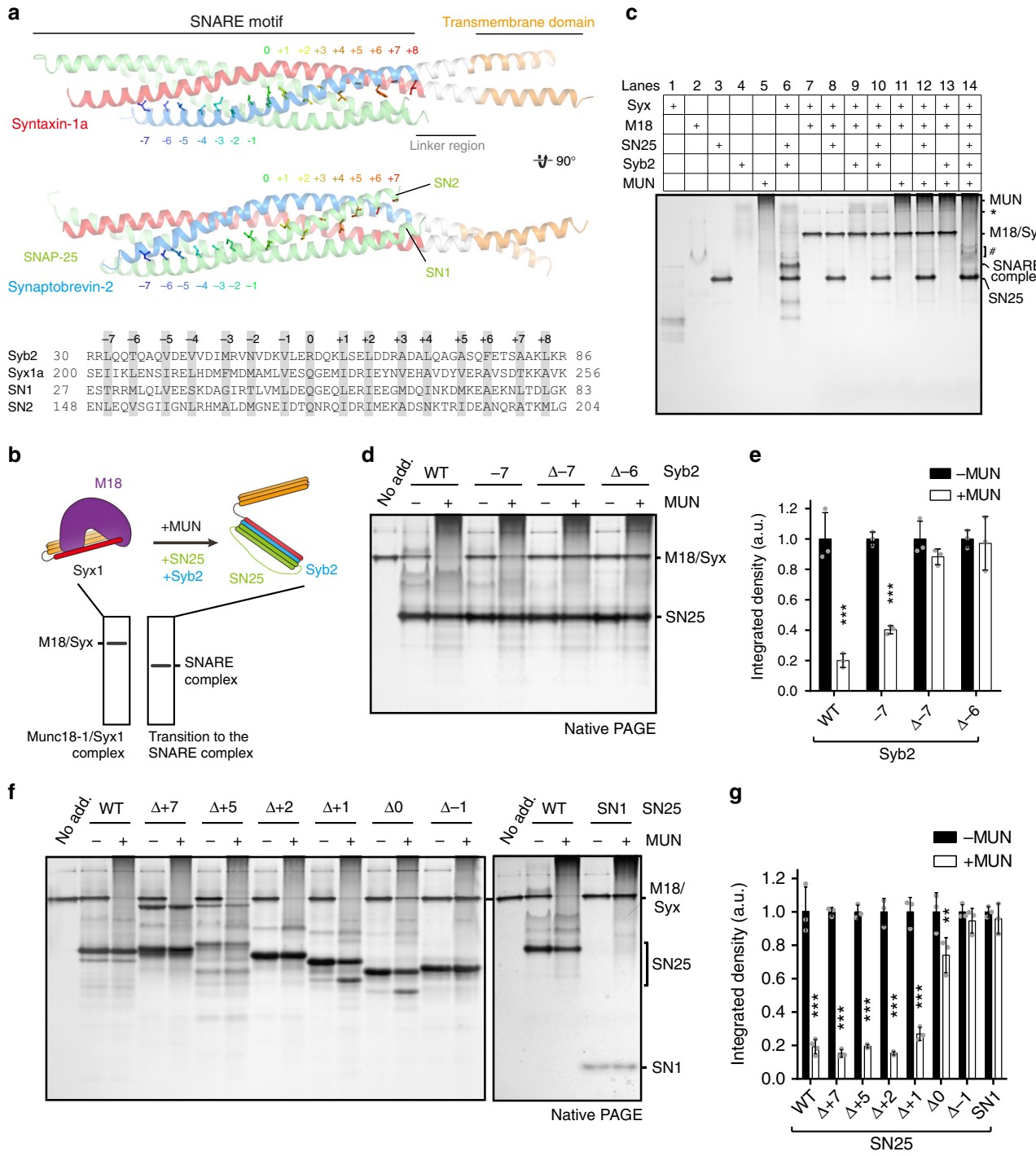

and function prior to SNARE zippering[17,18,29,30], we accordingly exploited SN25 as an indicator to address this issue. A series of truncations at different SN25 C-terminal layers (+7 to −1) were created, referred to as SN25$^{\Delta+7}$, SN25$^{\Delta+5}$, SN25$^{\Delta+2}$, SN25$^{\Delta+1}$, SN25$^{\Delta 0}$, and SN25$^{\Delta-1}$ (Supplementary Table 1). Intriguingly, SN25$^{\Delta+7}$, SN25$^{\Delta+5}$, SN25$^{\Delta+2}$, and SN25$^{\Delta+1}$ supported the transition as effectively as SN25 (Fig. 1f, g). However, SN25$^{\Delta 0}$ and SN25$^{\Delta-1}$ strongly impaired the transition, similar to a negative control (SN1) (Fig. 1f, g), implying that the N- to C-zippering of the SNAREs must break through the central layer 0 to accomplish the transition. In conclusion, in addition to Munc13-1 catalysis, a half-zippered SNARE assembly (from

layer −7 to layer 0) is required for the transition from the Munc18-1/Syx1 complex to the SNARE complex.

**N- and C-terminal parts of Syb2 are required for transition.** We also generated a series of Syb2 C-terminal truncations, referred to as Syb2$^{\Delta+8}$, Syb2$^{\Delta+7}$, Syb2$^{\Delta+6}$, Syb2$^{\Delta+5}$, and Syb2$^{\Delta+4}$ (Supplementary Table 1), to explore their potential defects in MUN-catalyzed transition to the SNARE complex. To our surprise, unlike the SN25 C-terminal truncations, these Syb2 C-terminal truncations all failed to support the transition (Fig. 2a, b), suggesting that Syb2 functions in the transition in a

**Fig. 1** A half-zippered SNARE assembly gates the transition. **a** Crystal structure of the SNARE complex (PDB entry: 3HD7). The lower panel displays the sequence of the SNARE motifs of Syx1, SN25, and Syb2. Hydrophobic binding layers −7 to +8 are indicated by rainbow-colored sticks in the upper panel and shaded in gray in the lower panel. **b** Scheme of the native PAGE assay for monitoring MUN-catalyzed transition from the Munc18-1/Syx1 complex to the SNARE complex. The Munc18-1/Syx1 complex (2 μM) displays a sharp band at the top of the gel; upon the addition of the MUN domain (30 μM), SN25 (10 μM), and Syb2 (10 μM), this band disappears with the formation of the SNARE complex. **c** Standard examples of the native PAGE assay. The Munc13-1 MUN domain, Munc18-1, or free Syb2 show smeared band; free Syx1 displays multi-bands that likely represent different assembly/aggregation states; SN25 shows a strong and clear band; Syx1 bound to Munc18-1 exhibits a strong and clear band. * indicates a putative non-productive aggregation of the Munc18-1/Syx1 complex. # indicates putative SNARE assembly intermediates that coexist with the actual ternary SNARE complex (lane 14). Disappearance of the Munc18-1/Syx1 complex can only be detected when all components are included (lane 14). Lane numbers are indicated at the top of the chart. **d** Effects of Syb2 N-terminal mutations or truncations on MUN-catalyzed transition from the Munc18-1/Syx1 complex to the SNARE complex using native PAGE. The representative gel displayed is one of three replicates. **e** Quantification of **d**. Integrated density represents the normalized integrated gray level of each assessed Munc18-1/Syx1 band. Data are presented as the means ± SD, $n = 3$, two-tailed $t$ test, ***$p < 0.001$. **f** Effects of SN25 C-terminal truncations on MUN-catalyzed transition from the Munc18-1/Syx1 complex to the SNARE complex using native PAGE. The representative gel displayed is one of three replicates. **g** Quantification of **f**. Integrated density represents the normalized integrated gray level of each assessed Munc18-1/Syx1 band. Data are presented as the means ± SD, $n = 3$, two-tailed $t$ test, **$p < 0.01$; ***$p < 0.001$. Source data are provided as a Source Data file

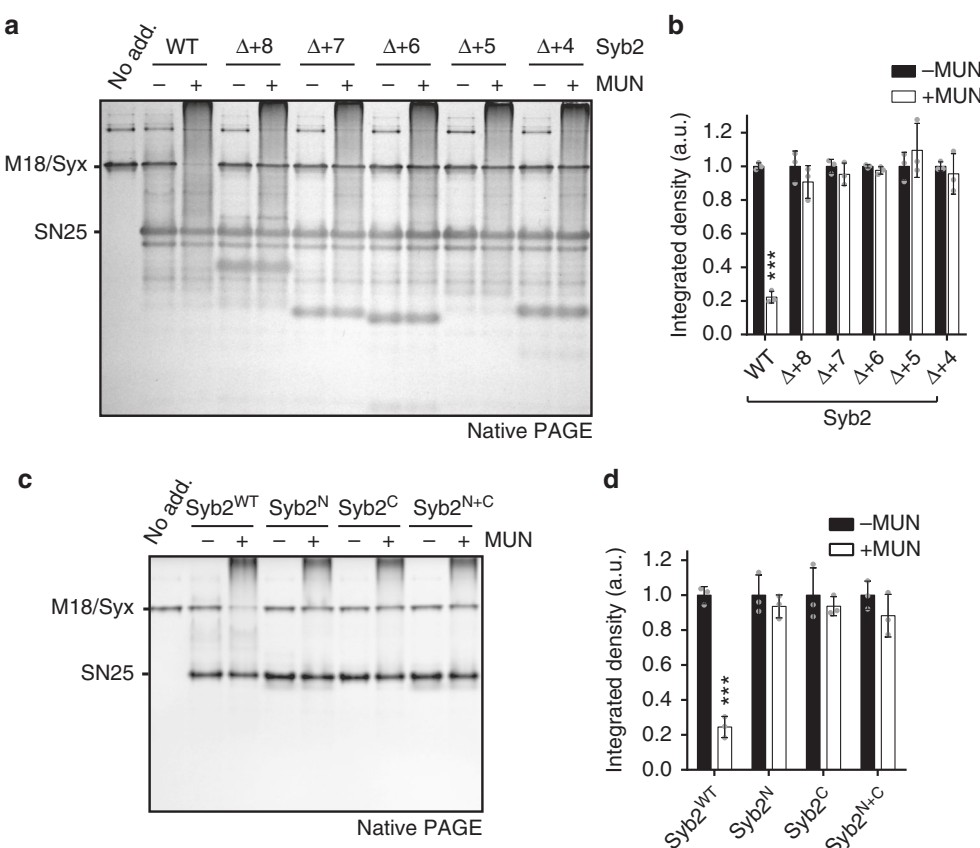

**Fig. 2** N- and C-portions of Syb2 are both required for the transition. **a** Effects of Syb2 C-terminal truncations on MUN-catalyzed transition from the Munc18-1/Syx1 complex to the SNARE complex using native PAGE. The representative gel displayed is from one of three replicates. **b** Quantification of **a**. Integrated density represents the normalized integrated gray level of each assessed Munc18-1/Syx1 band. Data are presented as the means ± SD, $n = 3$, two-tailed $t$ test, ***$p < 0.001$. **c** Functional analysis of the N- and/or C-portion of Syb2 in MUN-catalyzed transition from the Munc18-1/Syx1 complex to the SNARE complex using native PAGE. The representative gel displayed is from one of three replicates. **d** Quantification of **c**. Integrated density represents the normalized integrated gray level of each assessed Munc18-1/Syx1 band. Data are presented as the means ± SD, $n = 3$, two-tailed $t$ test, ***$p < 0.001$. Source data are provided as a Source Data file

manner distinct from that of SN25. Note that these C-terminal truncations do not influence SNARE N-terminal nucleation and zippering, as they displaced Syb2$^{49–96}$ from the ΔN-complex as effectively as Syb2 (Supplementary Fig. 2b).

As expected, neither Syb2$^N$ (residues 29–59, layers −7 to 0) nor Syb2$^C$ (residues 60–96, layers +1 to +8, plus the LR) (Supplementary Table 1) was able to support the transition when added separately (Fig. 2c, d). Intriguingly, the simultaneous addition of Syb2$^N$ and Syb2$^C$ (referred to as Syb2$^{N+C}$) also failed to support the transition (Fig. 2c, d), even though Syb2$^{N+C}$ preserved the ability to form a ternary complex with Syx1 and SN25 (Supplementary Fig. 3). The effectiveness of only the uninterrupted protein suggests that Syb2 requires its N- and C-portions to be tightly coupled in the transition.

**Crystal structure of the Syb2/MUN complex**. Intriguingly, the observation that even Syb2$^{\Delta+8}$ abrogated the transition (Fig. 2a, b) implies that the membrane-proximal LR (residues 84–96), outside the SNARE motif (Fig. 1a), is also indispensable for MUN-catalyzed transition from the Munc18-1/Syx1 complex to the SNARE complex. This observation, together with a recently reported interaction between the MUN domain and Syb2[30,32], led us to investigate whether the LR of Syb2 is associated with the interaction and function of the MUN domain. We thus generated multiple mutations on Syb2 (residues 29–96) and explored their interactions with the MUN domain using glutathione-S-transferase (GST) pull-down experiments (Supplementary Table 1 and Fig. 3a). Residues R86/K87 and W89/W90 residues in the LR of Syb2 were found to be essential for MUN domain interaction

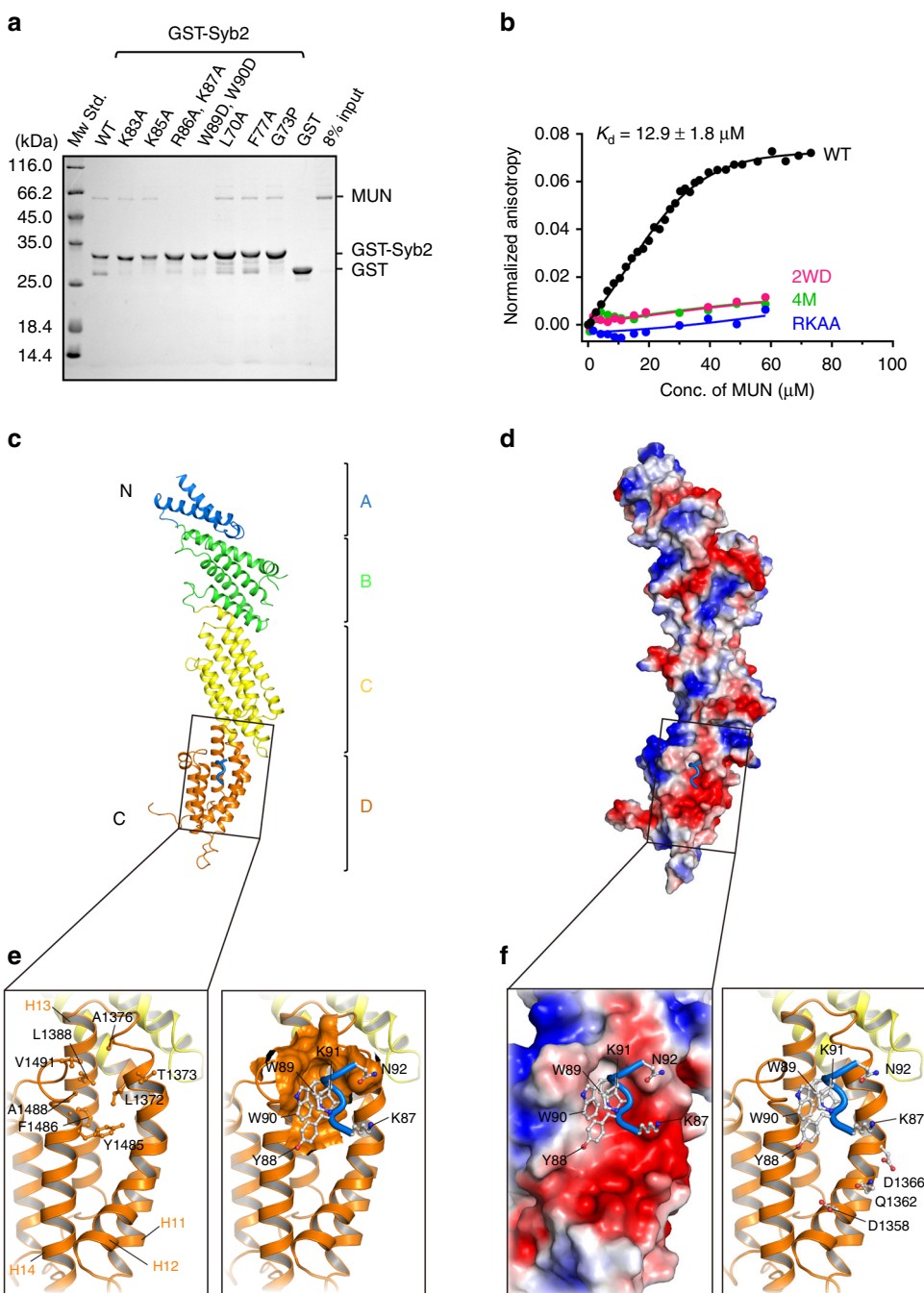

**Fig. 3** Crystal structure of the Syb2/MUN complex. **a** Screening of the MUN-binding sites on Syb2 (residues 29–96). Excess of the MUN protein or its mutants (3 µM) was added to GST-Syb2 (2 µM) in the GST pull-down assay. The representative gel displayed is from one of three replicates. **b** Determination of the disassociation constant between the MUN domain and Syb2 (residues 29–96) by fluorescence anisotropy. Data plots were fitted to the Hill equation, where the Hill coefficient (*n*) was fixed to 1. **c** Crystal structure of the Syb2/MUN complex. Subdomains of MUN are displayed by segmented color. Syb2 binds the MUN domain via a fragment of the Syb2 linker region (colored in blue). **d** Surface electron potential of the Syb2/MUN complex. The potential was scaled from −10 to 10kT/e, with red and blue denoting negative and positive potential, respectively. **e** Close-up view of the Syb2/MUN-binding interface on the MUN domain, showing a hydrophobic core formed by several residues on helices H12, H13, and H14 of subdomain D. **f** Close-up view of the negatively charged patch adjacent to the hydrophobic pocket bound to W89/W90. The right panel displays three potential residues (D1358, Q1362, and D1366) that are likely involved in Syb2 binding. Source data are provided as a Source Data file

(Fig. 3a). However, residues L70 and F77 in the SNARE motif of Syb2, predicted to bind Munc18-1 domain 3 (based on sequence alignment in Supplementary Fig. 4 and refs. [28,30]), did not participate in MUN domain interaction (Fig. 3a), suggesting that Syb2 interacts with Munc18-1 and the MUN domain via distinct binding sites. Syb2 (residues 29–96) was determined by fluorescence anisotropy to bind the MUN domain with a disassociation constant ($K_d$) of $12.9 \pm 1.8\,\mu M$, but Syb2$^{RKAA}$ (residues 29–96, R86A/K87A), Syb2$^{2WD}$ (residues 29–96, W89D/W90D), and Syb2$^{4M}$ (residues 29–96, R86A/K87A/W89D/W90D) all lost this binding ability (Fig. 3b).

To corroborate this interaction, we synthesized a Syb2 peptide (residues 75–93 of Syb2, containing residues R86/K87 and W89/W90, referred to as the LR). The LR/MUN complex was crystallized, and the complex structure was determined at 2.8 Å (Fig. 3c and Table 1). The C-terminal segment (residues 87–92) of LR was resolved in the complex structure (Fig. 3c–f). Residues W89 and W90 reside in a small hydrophobic pocket on subdomain D, formed by L1372, T1373, and A1376 on helix H12; L1388 on helix H13; and Y1485, F1486, A1488, and V1491 on helix H14 of the MUN domain (Fig. 3e). Unfortunately, the poor electron density for R86 and K87 impedes the observation of any direct interaction of these two charged residues with the MUN domain. This difficulty might arise because the high ionic strength applied in crystallization impairs the potential charge interactions (see Methods). Nevertheless, the backbone of K87 is observed to be positioned near a highly negatively charged patch on subdomain D, which is adjacent to the hydrophobic pocket bound to W89/W90 (Fig. 3f), indicating that both R86 and K87 are very likely to bind at this negatively charged patch under physiological conditions. In addition, the binding of LR to the MUN domain does not affect the global structure and conformation of the MUN domain (Supplementary Fig. 5).

Unlike the α-helical conformation of the Syb2 LR when bound to SN25 and Syx1, as observed in the crystal structure of the *cis*-SNARE complex (Fig. 1a), LR assumes a folded but not α-helical conformation when bound to the MUN domain (Fig. 3c–f), raising the question of whether the MUN domain maintains this interaction after SNARE complex formation. As detected by our GST pull-down experiment, the pre-assembled *cis*-SNARE complex showed reduced MUN domain interaction compared to that of isolated Syb2 (Supplementary Fig. 6). This binding result suggests that the MUN domain might dissociate from the LR of Syb2 upon the final zippering of the SNARE complex into the membrane, regardless of the likely maintained interaction between the MUN domain and the four-helix SNARE bundle, as detected by nuclear magnetic resonance experiment[19].

**Syb2/MUN interaction is essential for transition**. We next explored whether the Syb2/MUN interaction (mediated by LR) affects MUN-catalyzed transition from the Munc18-1/Syx1 complex to the SNARE complex. Consistent with the binding results (Fig. 3a, b), the transition was substantially influenced by Syb2$^{RKAA}$ and Syb2$^{2WD}$ and strongly abrogated by Syb2$^{4M}$ (Fig. 4a, b), demonstrating the functional importance of both hydrophobic and charge–charge interactions. In addition, we tested the above mutations by using a Munc18-1/Syx1$^{LE}$ system in which the transition to the SNARE complex can be achieved in the presence of SN25 and Syb2 with no need for the MUN domain[19,21]. In this system, Syb2$^{2WD}$, Syb2$^{RKAA}$, and Syb2$^{4M}$ all supported the transition as effectively as Syb2 even in the absence of the MUN domain (Supplementary Fig. 7). Hence, these data exclude functional interplay between Syb2 (LR) and Munc18-1, suggesting that the functional importance of the LR of Syb2 is related to its interaction with the MUN domain.

With in vitro fusion experiments using the Munc18-1/Syx1 complex as the starting component[20–22], we showed that both Syb2$^{RKAA}$ and Syb2$^{2WD}$ strongly reduced lipid mixing in the presence of the $C_1$-$C_2B$-MUN fragment, full-length synaptotagmin-1 (Syt1), SN25, and $Ca^{2+}$ (Fig. 4c, d), suggesting a critical role of R86/K87 and W89/W90 in membrane fusion. As expected, Syb2$^{4M}$ almost completely abrogated lipid mixing (Fig. 4c, d). These results highlight the essential role of the Syb2/MUN interaction in SNARE complex assembly and membrane fusion.

**Syb2/MUN interaction is essential for docking/priming**. To substantiate the physiological relevance of the Syb2/MUN interaction, we sought to identify those residues on the MUN domain that mediate the charge–charge interaction with residues R86 and K87 of Syb2. As shown in Fig. 3f, the negatively charged patch on subdomain D within the MUN domain is likely to be the potential target because this patch faces the backbone of residue K87, as observed in the complex structure. We mutated three negatively charged residues facing outward, that is, D1358K, Q1362K, and D1366K (Fig. 3f), and tested their functional defects in catalyzing the transition from the Munc18-1/Syx1 complex to the SNARE complex. Among these mutations, only the D1358K mutation showed impaired ability to catalyze the transition (Fig. 5a, b). As expected, this mutation strongly impaired the activity of the $C_1$-$C_2B$-MUN fragment to catalyze lipid mixing between liposomes reconstituted with the Munc18-1/Syx1 complex and liposomes reconstituted with Syb2 and Syt1 in the presence of SN25 and $Ca^{2+}$ (Fig. 5c, d), similar to the defect caused by the Syb2$^{RKAA}$ mutation (Fig. 4c, d).

We next explored the functional significance of D1358 in neurotransmitter release by using a knockdown (KD)-rescue approach in cultured mouse cortex neurons[21]. Consistent with the previous data[21], expression of the $C_1$-$C_2B$-MUN fragment in Munc13-1-deficient neurons was able to rescue both the spontaneous mini inhibitory postsynaptic current (mIPSC) frequency and the action potential-evoked IPSC (evoked IPSC) amplitude and charge transfer (Fig. 5e, f). However, the $C_1$-$C_2B$-MUN fragment bearing the D1358K mutation strongly impaired

### Table 1 Data collection and refinement statistics

|  | LR/MUN |
| --- | --- |
| Data collection |  |
| Space group | $P2_12_12$ |
| Cell dimensions |  |
| *a, b, c* (Å) | 114.48, 270.97, 47.915 |
| *α, β, γ* (°) | 90, 90, 90 |
| Resolution (Å) | 50–2.8 (2.85–2.80) |
| $R_{merge}$ | 0.095 (0.666) |
| $I/\sigma I$ | 24.7 (1.9) |
| Completeness (%) | 99.9 (100) |
| Redundancy | 6.4 (6.6) |
| No. of reflections | 37,919 |
| Refinement |  |
| Resolution (Å) | 42.0–2.8 |
| $R_{work}/R_{free}$ | 0.207/0.239 |
| No. of atoms |  |
| Protein | 4260 |
| Peptide | 68 |
| *B*-factor (Å$^2$) |  |
| Protein | 87.9 |
| Peptide | 132.3 |
| R.m.s deviations |  |
| Bond lengths (Å) | 0.004 |
| Bond angles (°) | 0.883 |

Values within parentheses are for highest-resolution shell

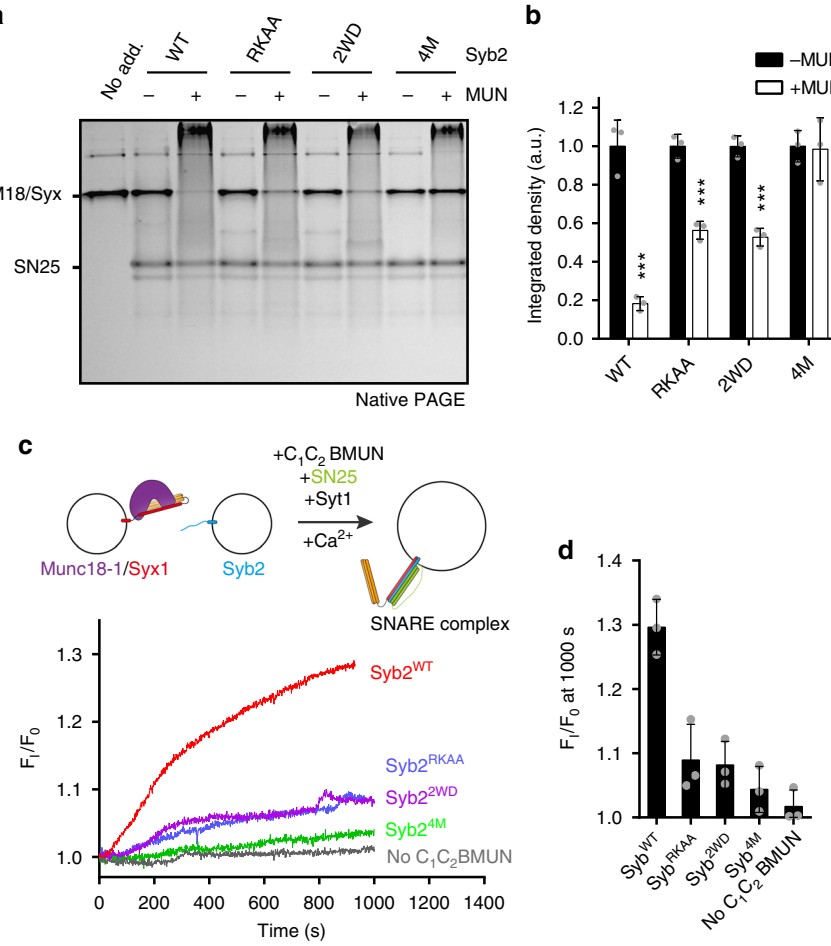

**Fig. 4** Syb2/MUN interaction is crucial for the transition as well as membrane fusion. **a** Functional analysis of Syb2 mutations in MUN-catalyzed transition from the Munc18-1/Syx1 complex to the SNARE complex. The RKAA, 2WD, and 4W mutations of Syb2 (residues 29–96) that disrupt the Syb2/MUN interaction strongly impaired the transition detected by native PAGE. The representative gel displayed is from one of three replicates. **b** Quantification of **a**. Integrated density represents the normalized integrated gray level of each assessed Munc18-1/Syx1 band. Data are presented as the means ± SD, $n = 3$, two-tailed $t$ test, ***$p < 0.001$. **c** Fusion between liposomes reconstituted with the Munc18-1/Syx1 (full-length) complex and liposomes reconstituted with Syb2 (full-length) and Syt1 in the presence of the $C_1$-$C_2B$-MUN fragment, SN25, and 1 mM $Ca^{2+}$. The RKAA, 2WD, and 4W mutations of Syb2 (residues 29–96) strongly impaired the fusion. Representative traces displayed are from one of three replicates. The liposome fusion assay is illustrated at the top of the chart. **d** Quantification of **c**. Data are presented as the means ± SD, $n = 3$. Source data are provided as a Source Data file

both mIPSCs and evoked IPSCs (Fig. 5e, f). Upon characterizing the size of the readily releasable pool (RRP)[34], we observed that the D1358K mutation led to a great reduction in RRP size (Fig. 5g), indicating that the D1358K mutation leads to defective docking or priming of exocytosis. These results suggest that the Syb2/MUN interaction plays an important role in docking/priming of exocytosis.

**Syb2/MUN interaction enables binding of Syb2 to Munc18-1.** Consistent with the Vps33/Nyv1 model[28], an interaction between the SNARE motif of Syb2 and Munc18-1 domain 3 was recently identified[29,30]. In support of this notion, we observed that mutating F77 of Syb2 (F77A, layer +6 of the SNARE motif, Fig. 1a), which likely impairs its interaction with Munc18-1 domain 3 but not the MUN domain (Fig. 3a and Supplementary Fig. 4), completely abrogated MUN-catalyzed transition from the Munc18-1/Syx1 complex to the SNARE complex (Fig. 6a, b), suggesting that binding of the SNARE motif of Syb2 to Munc18-1 domain 3 is also required for the transition. Our GST pull-down assay combined with immunoblotting detected a very weak interaction between Syb2 and the Munc18-1/Syx1 complex, but this interaction was not affected by the F77A mutation (Fig. 6c).

Intriguingly, inclusion of the MUN domain enhanced this interaction and rendered the interaction more sensitive to the F77A mutation (Fig. 6c). These results suggest that the MUN domain enables a more specific interaction between the SNARE motif of Syb2 and the Munc18-1/Syx1 complex.

Does this role of the MUN domain depend on the Syb2/MUN interaction? Strikingly, the Syb2[4M] mutation that disrupts the Syb2/MUN interaction caused the failure of the MUN domain to promote the interaction between Syb2 and the Munc18-1/Syx1 complex (Fig. 6c). These data, together with a Munc18-1/Syx1/MUN interaction identified previously[21,22], led us to believe that the MUN domain is likely to bring Syb2 in close proximity to the Munc18-1/Syx1 complex and thereby render the Syb2 SNARE motif more accessible to Munc18-1 domain 3 interaction (Supplementary Fig. 8). Hence, it is conceivable that Munc18-1 in coordination with Munc13-1 serves as a functional template to prime Syb2 and Syx1 for proper register.

**Syb2/MUN interaction is able to promote membrane association.** In addition to the Syb2/MUN interaction, the $C_1$-$C_2B$ domain that precedes the MUN domain in Munc13-1 binds DAG and PIP2 on the plasma membrane[20]. Thus, the $C_1$-$C_2B$-MUN

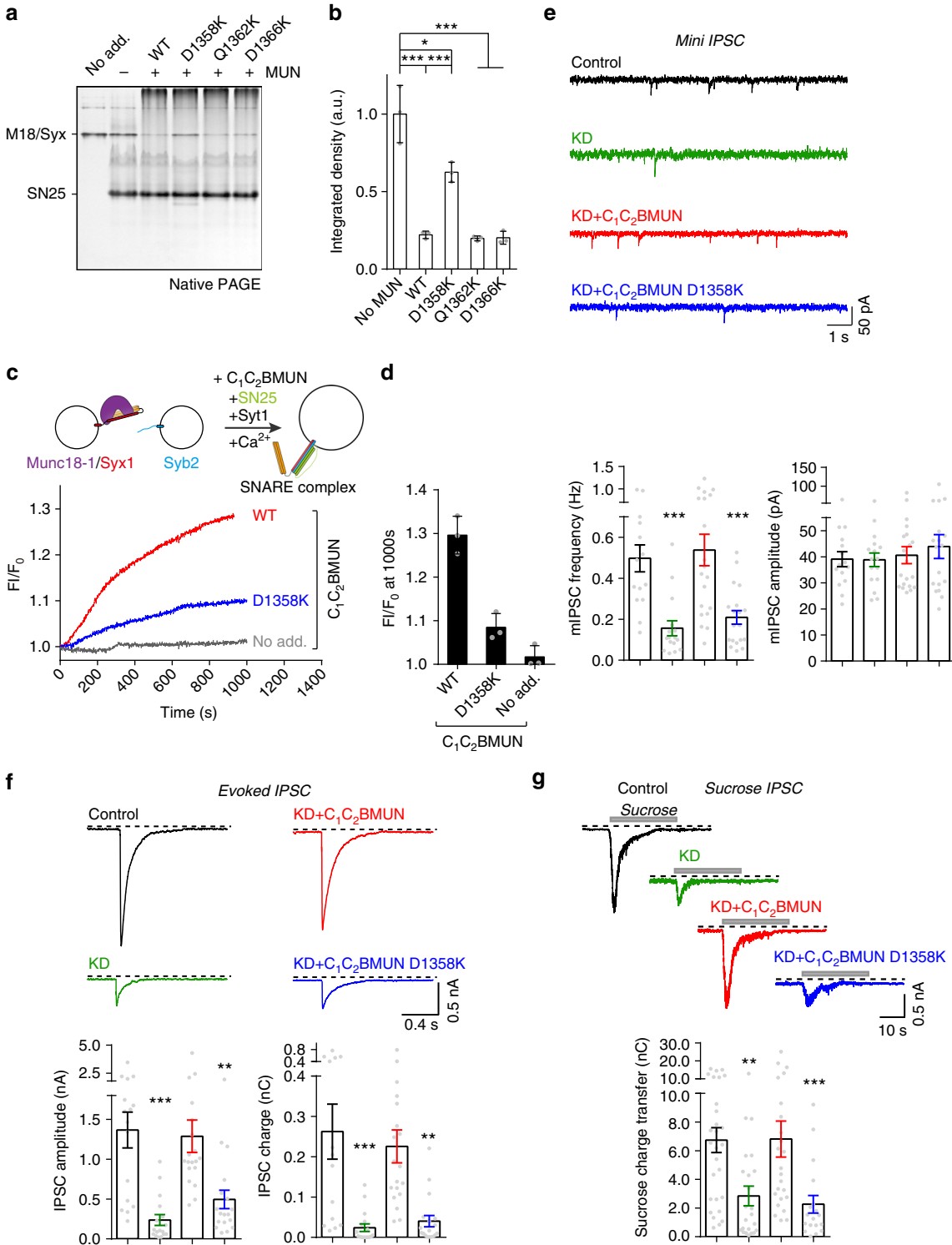

fragment would be expected to promote association between membranes bearing full-length Syb2 and membranes bearing DAG/PIP2. To test this hypothesis, we employed a single-vesicle tethering assay as previously described[35,36]. Artificial plain vesicles bearing DAG/PIP2 (termed PM-vesicles) were immobilized on polyethylene glycol-passivated glass surfaces to mimic the plasma membrane, whereas vesicles bearing reconstituted full-length Syb2 (termed SV-vesicles) were prepared to mimic synaptic vesicles (Fig. 7a). Control experiments ensured that the density of the immobilized PM-vesicles remained constant in all

trials (Supplementary Fig. 9). The $C_1$-$C_2$B-MUN fragment was added together with SV-vesicles to the supported PM-vesicles for a 30-min incubation period. After extensive buffer washing, SV-vesicles robustly tethered onto the supported PM-vesicles were detected by an enhanced fluorescence signal (Fig. 7b, i), supporting our hypothesis that the $C_1$-$C_2$B-MUN fragment promotes association between SV- and PM-vesicles. In contrast, the mutations that disrupt the Syb2/MUN interaction (Syb2$^{RKAA}$, Syb2$^{2WD}$, and Syb2$^{4M}$) showed marked defect in supporting this membrane association (Fig. 7c–e, i), and a similar defect was

**Fig. 5** Syb2/MUN interaction is essential for synaptic vesicle docking/priming. **a** Functional analysis of mutations in the negatively charged patch of the Munc13-1 MUN domain in the transition from the Munc18-1/Syx1 complex to the SNARE complex using native PAGE. The representative gel displayed is from one of three replicates. **b** Quantification of **a**. Integrated density represents the normalized integrated gray level of each assessed Munc18-1/Syx1 band. Data are presented as the means ± SD, n = 3, two-tailed t test, *p < 0.05; ***p < 0.001. **c** Fusion between liposomes reconstituted with the Munc18-1/Syx1 complex and liposomes reconstituted with Syb2 and Syt1 in the presence of the $C_1$-$C_2B$-MUN fragment, SN25, and 1 mM $Ca^{2+}$. Representative traces displayed are from one of three replicates. The liposome fusion assay is illustrated at the top of the chart. **d** Quantification of **c**. Data are presented as the means ± SD, n = 3. **e** Sample traces (top) and statistical summary (bottom) of mini IPSCs recorded in neuronal cultures that were infected with a control lentivirus (Control) (n = 14) or a lentivirus expressing only Munc13-1 shRNAs (KD) (n = 16) or Munc13-1 shRNAs plus either the $C_1$-$C_2B$-MUN fragment (KD/$C_1$-$C_2B$-MUN) (n = 21) or the D1358K mutation (KD/$C_1$-$C_2B$-MUN D1358K) (n = 12), respectively. Recorded cells are from three independent litters of mice. **f** Sample traces (top) and statistical summary (bottom) of action potential-evoked IPSCs recorded in neuronal cultures that were infected with a control lentivirus (Control) (n = 16) or a lentivirus expressing only Munc13-1 shRNAs (KD) (n = 17) or Munc13-1 shRNAs plus either the $C_1$-$C_2B$-MUN fragment (KD/$C_1$-$C_2B$-MUN) (n = 19) or the D1358K mutation (KD/$C_1$-$C_2B$-MUN D1358K) (n = 18), respectively. Recorded cells are from three independent litters of mice. **g** Sample traces (top) and statistical summary (bottom) of IPSCs evoked by 0.5 M sucrose recorded in neuronal cultures that were infected with a control lentivirus (Control) (n = 28) or a lentivirus expressing only Munc13-1 shRNAs (KD) (n = 24) or Munc13-1 shRNAs plus either the $C_1$-$C_2B$-MUN fragment (KD/$C_1$-$C_2B$-MUN) (n = 24) or the D1358K mutation (KD/$C_1$-$C_2B$-MUN D1358K) (n = 18), respectively. Recorded cells are from four independent litters of mice. **p < 0.01; ***p < 0.001, two-tailed t test. Source data are provided as a Source Data file

observed for the D1358K mutation (Fig. 7g, j). These results imply that the Syb2/MUN interaction might play a role in membrane association. Notably, deletion of the $C_1$-$C_2B$-MUN fragment caused a stronger defect in membrane association than deletion of Syb2 (Fig. 7f, h, j), implying the involvement of an additional MUN/lipid interaction[37]. Altogether, these results suggest that the $C_1$-$C_2B$-MUN fragment is able to align Syb2-embedded synaptic vesicles precisely to PIP2-enriched active zones on the plasma membrane, thus enhancing the binding reactivity between Syb2 and the Munc18-1/Syx1 complex (Fig. 8a).

## Discussion

Recent studies illustrated a Munc18–Munc13 route to SNARE complex assembly that enables the exquisite regulation of synaptic exocytosis. Central to this route is the transition from the Munc18-1/Syx1 complex to the SNARE complex, which is regulated by Munc13-1, SN25, and Syb2[19,20]. Increasing evidence has revealed that Munc13-1 promotes the transition via the binding of its MUN domain to the Munc18-1/Syx1 complex[21,22], and Munc18-1 promotes the transition via the interaction of its domain 3 with Syb2[28–30]. Nevertheless, a coherent picture of the sequential and cooperative actions among Munc18-1, Munc13-1, and the SNAREs in the transition has not emerged, hindering our understanding of the exquisite regulation of exocytosis. Here, we have identified a specific interaction between the Munc13-1 MUN domain and the Syb2 LR. Using combined in vitro and in vivo approaches, we demonstrated that this interaction plays an important function in (I) the transition from the Munc18-1/Syx1 complex to the SNARE complex; (II) Munc13-catalyzed membrane fusion; (III) membrane association; and (IV) docking/priming of exocytosis.

Interactions between SNAREs and CATCHRs are of great significance in diverse membrane trafficking systems[38]. For example, the Dsl1 complex interacts with the target SNAREs Ufe1 and Sec20 at the endoplasmic reticulum and the coat protein complex I vesicles, thus promoting SNARE complex assembly by capturing the vesicles[39]; the exocyst complex simultaneously binds to SNAREs and PIP2 and promotes Sso1/Sec9 complex assembly[40,41]; the Golgi-associated retrograde protein complex interacts with Tlg1, thus maintaining protein sorting in the late Golgi phase[42]; the conserved oligomeric Golgi complex orchestrates intra-Golgi vesicular transport via interaction with Syx5/Sed5[43]; and the homotypic fusion and vacuolar protein sorting (HOPS) complex binds to assembled vacuolar SNARE complexes to facilitate vacuole-to-vacuole fusion[44,45]. As the MUN domain is structurally similar to the CATCHRs[12,13], our identification of

a functionally relevant interaction between the Munc13-1 MUN domain and the LR of Syb2 ideally fills a vacancy in the enigmatic interactive pattern between neuronal CATCHRs and SNAREs.

The Syb2/MUN interaction mediated by the LR of Syb2 is compatible with recent biochemical data showing that the SNARE motif of Syb2 is specific for Munc18-1 domain 3 binding[29,30], suggesting that isolated Syb2 inherently harbors non-overlapping binding sites for Munc18-1 and the MUN domain. In addition, recent crystal structures of Vps33 bound to Vam3 and to Nyv1 suggest a templating role of Munc18-1 that depends on simultaneous binding with Syx1 and Syb2[28]. Furthermore, the MUN domain likewise comprises two discrete SNARE-binding sites[21,22]: one for the Syb2 LR identified in this work (D1358 in subdomain D) (Figs. 3 and 5), and the other for Syx1 bound to Munc18-1 (the NF pocket at the junction of subdomain B and C) identified previously[21,22]. These multiple and reciprocal interactions may cooperate to construct a quadruple MUN/Syb2/Munc18-1/Syx1 complex, which likely represents an intermediate in Munc18–Munc13 route to SNARE complex assembly (Supplementary Fig. 8 and Fig. 8b). The assembly of this quadruple complex was strongly supported by our finding that the MUN domain mediates the interaction between Syb2 and the Munc18-1/Syx1 complex (Fig. 6c). It is also very likely that the MUN domain together with Syb2 induces the extension of Munc18-1 domain 3, as the bent structure of domain 3 adopted in the Munc18-1/Syx1 complex is incompatible with Syb2 binding and SNARE complex assembly[28,30]. Consistent with this possibility, the HOPS subunit Vps16 was found to be required for the Vps33/Nyv1 complex crystallization[28,46], probably owing to a MUN-like function of Vps16 in stabilizing the Vps33/Nyv1 interaction. Moreover, the observation that the $C_1$-$C_2B$-MUN fragment allowed the anchoring of Syb2-vesicles to the supported bilayers containing DAG/PIP2 leads to the notion that the Syb2/MUN interaction might pre-align synaptic vesicles with the plasma membrane, thus enhancing the binding reactivity between Syb2 and the Munc18-1/Syx1 complex (Fig. 8a).

Following the assembly of the MUN/Syb2/Munc18-1/Syx1 complex, the entry of SN25 is expected to ensure simultaneous SNARE N-terminal nucleation and zippering, hindering the formation of a 2:1 "dead-end" Syx1/SN25 complex (Fig. 8c). Interestingly, recent data revealed that Munc18-1 and Munc13-1 work together to properly assemble the SNARE four-helical bundle by promoting both the proper Syx1/Syb2 and Syx1/SN25 subconfigurations within the ternary SNARE complex[32]. Chaperoned by Munc18-1 and the MUN domain, a half-zippered SNARE assembly, from layer −7 to the central layer 0, was observed to be essential for the transition from the Munc18-1/Syx1

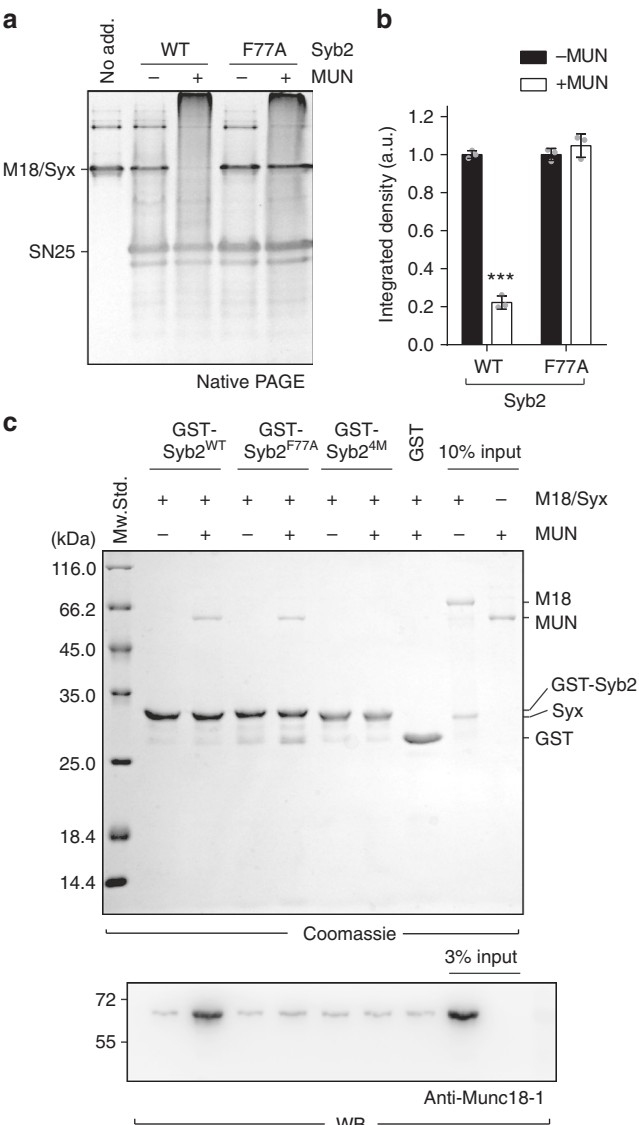

**Fig. 6** The MUN domain enables specific binding of Syb2 to the Munc18-1/Syx1 complex. **a** The F77A mutation of Syb2 severely impairs MUN-catalyzed transition from the Munc18-1/Syx1 complex to the SNARE complex. The representative gel displayed is from one of three replicates. **b** Quantification of **a**. Integrated density represents the normalized integrated gray level of each assessed Munc18-1/Syx1 band. Data are presented as the means ± SD, $n = 3$, two-tailed $t$ test, ***$p < 0.001$. **c** Binding of Syb2 and its mutations to the Munc18-1/Syx1 complex in the absence or presence of the MUN domain detected by GST pull-down assay combined with immunoblotting. The MUN domain enables an interaction between Syb2 and the Munc18-1/Syx1 complex, and this interaction was able to be disrupted by the F77A or 4M mutation. The representative gels displayed are from one of three replicates. Source data are provided as a Source Data file

complex to the SNARE complex (Fig. 1). This notion is consistent with a recent finding that a β-hairpin within Munc18-1, which interacts with the central layer of the Syx1 SNARE domain, gates the opening of Syx1[47]. In addition, this half-zippered SNARE assembly may allow complexin and/or Syt1 binding and regulation[48–50], thus leading to an irreversible C-terminal assembly of the SNARE complex into the membrane (Fig. 8d).

The observation that the MUN domain prefers to bind a non-α-helical state of the Syb2 LR (Supplementary Fig. 6) indicates that the MUN domain might dissociate from the Syb2 LR upon

the full zippering of the SNARE complex into the membranes. As a consequence, the polybasic residues ahead of residues W89 and W90 in the Syb2 LR likely turn to establish a positive electrostatic potential at the membrane surface, which is proposed to locally reduce repulsive forces between the negative surface potentials of the opposing membranes, thereby facilitating Ca$^{2+}$-triggered membrane fusion[51–53].

Hence, these data provide compelling evidence that Munc18-1 and Munc13-1 cooperate to enhance binding specificity among the cognate SNAREs between synaptic vesicles and the pre-synaptic membrane, suggesting that Munc18-1 in coordination with Munc13-1 serves as a functional template that primes Syb2, Syx1, and SN25 to ensure proper SNARE assembly (Fig. 8b). This templating function of Munc18-1 and Munc13-1 in synaptic exocytosis may constitute a general function of SMs and CATCHRs in other intracellular membrane docking and fusion reactions.

## Methods

**Plasmids and protein purification.** The full-length rat Syb2 (residues 1–116), its cytoplasmic domain (residues 29–96), a series of C-terminal truncations and mutations Syb2$^{\Delta+8}$ (Δ84–96, residues 29–83), Syb2$^{\Delta+7}$ (Δ81–96, residues 29–80), Syb2$^{\Delta+6}$ (Δ77–96, residues 29–76), Syb2$^{\Delta+5}$ (Δ74–96, residues 29–73), Syb2$^{\Delta+4}$ (Δ70–96, residues 29-69), N-terminal truncations Syb2$^{\Delta-7}$ (Δ29–34, residues 35–96), Syb2$^{\Delta-6}$ (Δ29–38, residues 39–96), Syb2$^{-7}$ (residues 29–96, L32A), the N-terminal half (residues 29–59, Syb2$^N$), the C-terminal half (residues 60–96, Syb2$^C$), Syb2$^{49-96}$ (residues 49–96), and various Syb2 point mutations were all constructed into the pGEX-KG vector. Full-length human SN25 (with four native cysteines mutated to serines), its C-terminal truncations SN25$^{\Delta+7}$ (Δ199–206, residues 1–198), SN25$^{\Delta+5}$ (Δ192–206, residues 1–191), SN25$^{\Delta+2}$ (Δ181–206, residues 1–180), SN25$^{\Delta+1}$ (Δ178–206, residues 1–177), SN25$^{\Delta0}$ (Δ174–206, residues 1–173), SN25$^{\Delta-1}$ (Δ171–206, residues 1–170), rat Syt1 full-length (C74A, C75A, C77A, C79A, and C82A), and the cytoplasmic domain of rat Syx1a (residues 1–261) were cloned into the pET28a vector (Novagen). Human SN25a first SNARE domain (SN1, residues 1–83), rat Munc13-1 MUN domain (residues 933–1407, EF, 1453–1531), and its mutation MUN D1358K, Q1362K, and D1366K were cloned into the pGEX-KG vector. The co-expressed rat Munc18-1/Syx1 (residues 1–288), Munc18-1/Syx1 (residues 1–261), and Munc18-1/Syx1$^{LE}$ (residues 1–261, L165A, E166A) were constructed into the pETDuet-1 vector (Novagen). The proteins above were all expressed in *Escherichia coli* BL21 DE3 and purified as previously described[21,22,54]. For GST-tagged proteins, cell pellets from 1 L of culture were resuspended with 50 mM Na$_2$HPO$_4$-NaH$_2$PO$_4$, pH 7.6, 300 mM NaCl, 0.5% Triton X-100 (Sigma; lysis buffer A) supplied with 1 mM phenylmethanesulfonyl fluoride (PMSF, Amresco) and 5 mM 2-mercaptoethanol (2-ME, Amresco). Cells were broken using an AH-1500 Nano Homogenize Machine (ATS Engineering Inc.) at 1200 bar for three times at 4 °C. Cell lysates were centrifuged at 16,000 rpm in a JA-25.50 rotor (Beckman Coulter) at 4 °C. The supernatants were collected and mixed with 1 ml glutathione Sepharose 4B (GE Healthcare) affinity media. After 2 h rotation at 4 °C, the mixture were washed twice with lysis buffer A supplied with 2 mM 2-ME. Thrombin (Sigma) was used for removing GST fusion tag. In order to obtain GST-tagged proteins for pull-down assays, 10 mM glutathione (Biosharp) was applied into the elution buffer (50 mM Tris-Cl, pH 8.0, 300 mM NaCl). For hexa-histidine-tagged proteins, cell pellets from 1 L of culture were resuspended with 50 mM Tris-Cl, pH 8.0, 300 mM NaCl, 0.5% Triton X-100 (Sigma; lysis buffer B) supplied with 1 mM PMSF and 2 mM 2-ME. Cells were broken using an AH-1500 Nano Homogenize Machine (ATS Engineering Inc.) at 1200 bar for three times at 4 °C. Cell lysates were centrifuged at 16,000 rpm in a JA-25.50 rotor (Beckman Coulter) at 4 °C. The supernatants were collected and mixed with 1 ml Nickel-NTA agarose (Qiagen) affinity media. After 2 h rotation at 4 °C, the mixture were washed twice with lysis buffer B supplied with 1 mM 2-ME and 30 mM imidazole followed by an additional wash step with Triton X-100-free lysis buffer B supplied with 1 mM 2-ME and 30 mM imidazole. Proteins were eluted with a buffer containing 20 mM Tris-Cl, pH 8.0, 150 mM NaCl, 10% (v/v) glycerol (buffer T) supplied with 300 mM imidazole, and 0.2 mM Tris(carboxyethyl)-phosphine (TCEP). For transmembrane proteins, the elution buffer contains additional 1% (w/v) 3-[(3-cholamidopropyl)dimethylammonio]-1-propanesulfonate (CHAPS, Amersco). The rat Munc13-1 C$_1$-C$_2$B-MUN fragment (residues 529–1407, EF, 1453–1531) and its D1358K mutation were cloned into pFastBac$^{TM}$HtB vector (Invitrogen). The proteins were expressed in Sf9 insect cells as previously described[20]. The construct of pFastBac$^{TM}$HtB-Munc13-1 C$_1$-C$_2$B-MUN was used to generate a baculovirus using the Bac-to-Bac system (Invitrogen). Sf9 insect cells were infected with the baculovirus, harvested about 68–72 h post infection, and resuspended in lysis buffer B supplied with 1 mM PMSF, 2 mM 2-ME, and 10 mM imidazole. Cells were broken using an AH-1500 Nano Homogenize Machine (ATS Engineering Inc.) at 800 bar for three times at 4 °C. Cell lysates were centrifuged at 16,000 rpm

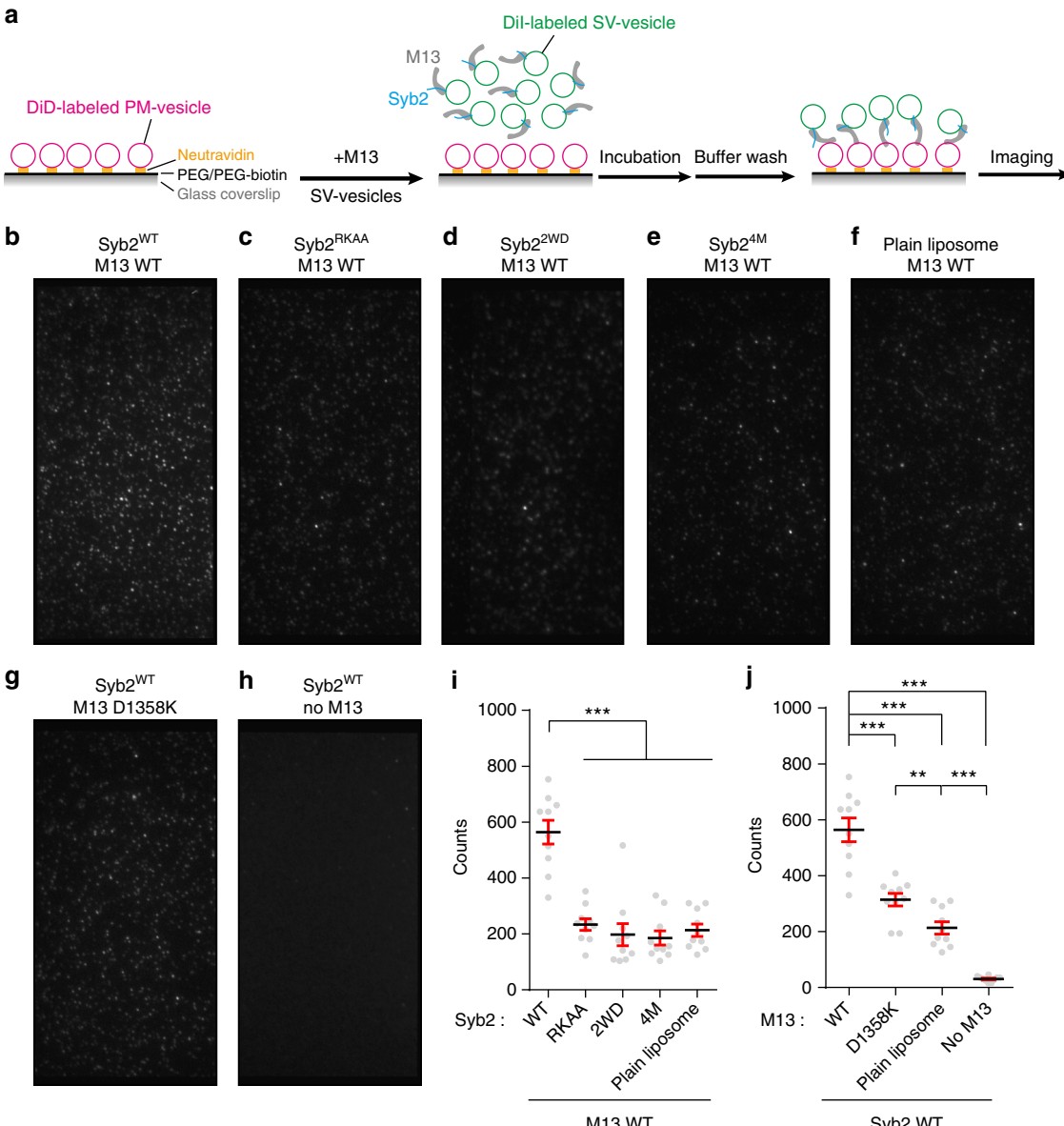

**Fig. 7** Syb2/MUN interaction promotes membrane association. **a** Illustration of the single-vesicle tethering assay. Glass surfaces were modified by PEG and biotin-PEG mixtures. DiD-labeled PM-vesicles containing 0.5% biotin-PE were first immobilized on a surface treated with neutravidin. After extensive washing, 1 μM Munc13-1 $C_1$-$C_2$B-MUN fragment (M13) and 40 μM (total lipids) DiI-labeled SV-vesicles were added and incubated for 30 min at 30 °C. Finally, another washing step was performed before imaging. **b** Representative channel image using Syb2$^{WT}$-bearing SV-vesicles and wild-type Munc13-1 $C_1$-$C_2$B-MUN fragment (M13). **c** Example channel image using Syb2$^{RKAA}$-bearing SV-vesicles and wild-type Munc13-1 $C_1$-$C_2$B-MUN fragment (M13). **d** Example channel image using Syb2$^{2WD}$-bearing SV-vesicles and wild-type Munc13-1 $C_1$-$C_2$B-MUN fragment (M13). **e** Example channel image using Syb2$^{4M}$-bearing SV-vesicles and wild-type Munc13-1 $C_1$-$C_2$B-MUN fragment (M13). **f** Example channel image using plain SV-vesicles and wild -type Munc13-1 $C_1$-$C_2$B-MUN fragment (M13). **g** Example channel image using Syb2$^{WT}$-bearing SV-vesicle and Munc13-1 $C_1$-$C_2$B-MUN fragment (M13) D1358K mutant. **h** Example channel image using Syb2$^{WT}$-bearing SV-vesicles only. **i** Quantification of the results in **b**–**f**. **j** Quantification of the results in **b**, **f**–**h**. Data are presented as the means ± SEM with dots showing individual single-vesicle counts from 10 randomly chosen frames ($n = 10$). $^{**}p < 0.01$; $^{***}p < 0.001$, two-tailed $t$ test. Source data are provided as a Source Data file

in a JA-25.50 rotor (Beckman Coulter) at 4 °C. The supernatants were collected and mixed with 1 ml Nickel-NTA agarose (Qiagen) affinity media. After 1 h rotation at 4 °C, the mixture were washed twice with lysis buffer B supplied with 1 mM 2-ME and 30 mM imidazole, followed by an additional wash step with Triton X-100-free lysis buffer B supplied with 1 mM 2-ME and 30 mM imidazole. Proteins were finally eluted with buffer T supplied with 300 mM imidazole and 0.2 mM TCEP. All of the eluted proteins were loaded into Superdex 200 pg or Superdex 75 pg size exclusion chromatography (GE Healthcare) to remove aggregates and potential contaminants. For Munc13-1 MUN domain, its mutations, and Syt1 full-length protein, additional ion exchange chromatography (Source Q or Source S, GE Healthcare) were applied to remove nucleotide contaminations.

**Native PAGE assays**. The reactions were performed in the non-denaturing (sodium dodecyl sulfate (SDS)-free) conditions as previously described[22]. Firstly, to detect the transition from the Munc18/Syx1 complex to the SNARE complex, 2 μM co-expressed Munc18-1/Syx1 (residues 1–261) complex was mixed with 10 μM SN25, 10 μM cytoplasmic domain of Syb2 (residues 29–96) and 30 μM MUN (residues 933–1407, EF, 1453–1531) or their mutations at 30 °C for 1 h. For SNARE complex assembly assay shown in Supplementary Fig. 3, 4 μM Syx1, 5 μM SN25, 10 μM Syb2$^{N}$ (residues 29–59, layers −7 to 0) and 10 μM Syb2$^{C}$ (residues 60–96, layers +1 to +8) were mixed at 30 °C for 1 h. The samples were loaded into the non-denaturing gel composed of 15% polyacrylamide in the separating gel (pH 8.4) and 5% in the stacking gel (pH 6.8), running in the native electrophoresis buffer containing 25 mM Tris-Cl, 250 mM glycine, pH 8.5, at 4 °C overnight. For

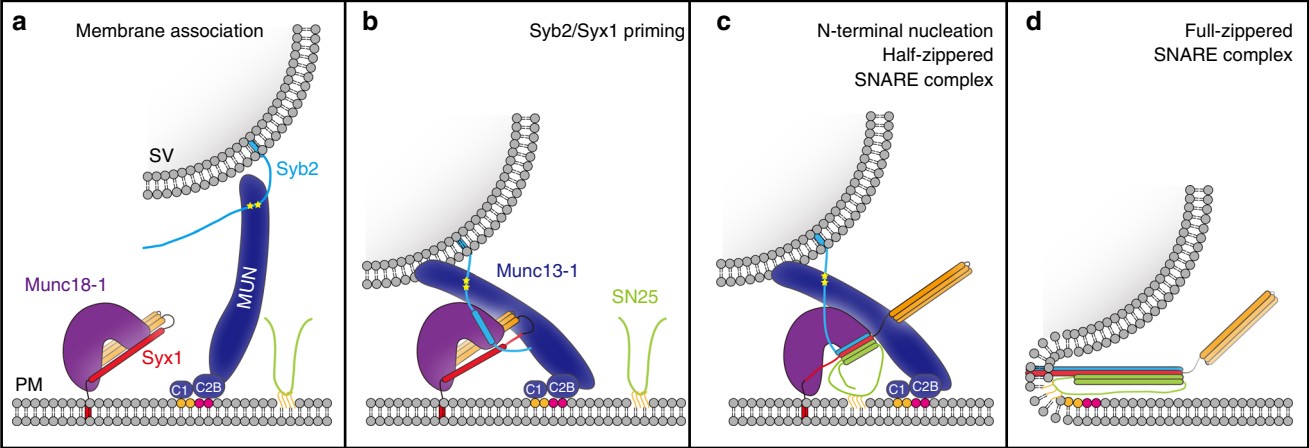

**Fig. 8** Working model of Munc18–Munc13 route to SNARE complex assembly. **a** The Syb2/MUN interaction and the $C_1$-$C_2B$/DAG-$PIP_2$ interaction might pre-align synaptic vesicles with the plasma membrane, thus enhancing the binding reactivity between Syb2 and the Munc18-1/Syx1 complex. Yellow pentagrams represent the hydrophobic (W89/W90) and charged (R86/K87) residues in the Syb2 linker region. Orange and magenta spheres indicate DAG and $PIP_2$, respectively. **b** The Syb2/MUN interaction, in coordination with the Munc18-1/Syx1/MUN interaction, allows binding of Syb2 to the extended conformation of domain 3 and thereby primes Syb2 and Syx1. **c** Munc18-1 and Munc13-1 proofread SNARE N-terminal nucleation with the entry of SN25. A half-zippered SNARE assembly releases Syx1 from Munc18-1 clamping, thus gating the transition to the SNARE complex. **d** Full zippering of the C-terminal region of the SNARE complex into the membrane leads to fusion of synaptic vesicles with the plasma membrane

immunoblotting experiment in Supplementary Fig. 1, Munc18-1 rabbit polyclonal antibody (Proteintech #11459-1-AP) (1:1000 dilution in use) and Syx1 mouse monoclonal antibody (Proteintech #66437-1-lg) (1:1000 dilution in use) were applied, respectively. Each of the reactions was repeated at least three times, with the quantification of the Munc18-1/Syx1 complex bands analyzed by Image J (NIH).

**Peptide**. Syb2 linker region peptide (sequence: SQFETSAAKLKRKYWWKNL) used for crystallization was synthesized and obtained from Scilight Biotechnology (Beijing, China) with a purity >99% as determined by mass spectrum.

**Crystallization and data collection**. The Munc13-1 MUN domain was crystallized by using hanging-drop vapor diffusion at 4 °C by mixing equal volumes of protein (10 mg/ml) and well solution containing 20% PEG 3350 (v/v), 0.1 M MES, pH 5.8–6.3, and 0.2 M $Mg(NO_3)_2$, which was the same to our previously work[22]. The synthesized Syb2 peptide (LR) was dissolved in a buffer containing 20 mM Tris-Cl, 150 mM NaCl, 10% glycerol (v/v), pH 8.5. After the crystals were grown well at 4 °C for 7 days, the peptide was soaked into the crystals at a 1:4 (MUN: peptide) molar ratio to grow for another 7 days at 4 °C. Before data collection, crystals were soaked in the reservoir solution with 30% glycerol and flash cooled in liquid nitrogen. All the diffraction data were collected at a wavelength of 0.9793 Å at 100 K on beamline BL19U1 of Shanghai Synchrotron Radiation Facility. The data were collected at 2.8 Å; the LR/MUN complex structure was solved by molecular replacement using MUN (PDB entry: 4Y21[22]) as a starting model. Molecular replacement was performed using the program Phaser[55] in the CCP4 suite; the model was manually improved in COOT[56].

**GST pull-down assay**. For detecting the interaction between MUN domain and Syb2, 2 µM GST-Syb2 (residues 29–96), 3 µM hexa-histidine-tagged-MUN domain protein (residues 933–1407, EF, 1453–1531) were mixed with 10 µl 50% (v/v) glutathione Sepharose 4B affinity media (GE Healthcare) to a final volume of 50 µl at 4 °C for 3 h. For detecting the binding between Syb2 and the Munc18-1/Syx1 (residues 1–261) complex in the absence/presence of the MUN domain, 2 µM GST-Syb2, or its mutants, 3 µM MUN (if applicable), 3 µM Munc18-1/Syx1 complex were mixed with 10 µl of 50% (v/v) glutathione Sepharose 4B affinity media (GE Healthcare) to a final volume of 50 µl at 4 °C for 3 h. For MUN domain interaction with Syb2 and cis-SNARE complex assay, purified GST-Syb2 (residues 29–96) was mixed with SN25 and Syx1 (residues 1–261) at 4 °C overnight to confirm full SNARE complex formation; then, 2 µM GST-Syb2 or GST-SNARE complex, 3 µM MUN were mixed with 10 µl of 50% (v/v) glutathione Sepharose 4B affinity media (GE Healthcare) to a final volume of 50 µl at 4 °C for 3 h. After gently shaking, the media-bound samples were washed by 25 mM HEPES-K+, pH 7.4, 150 mM KCl, and 10% glycerol (v/v) for three times, and the samples were boiled and analyzed by SDS-PAGE and western blot (if applicable). For Fig. 6c, Munc18-1 rabbit polyclonal antibody (Proteintech #11459-1-AP) was applied (1:1000 dilution in use). Uncropped western blot are shown in Supplementary Figure 10.

**Fluorescence anisotropy**. Fluorescence anisotropy assays were performed on PTI QM-40 fluorescence spectrophotometer, equipped with a set of polarizers, and with an excitation/emission wavelength at 485/513 nm. Basic procedures were described elsewhere[8,21,57]. For monitoring the binding $K_d$ between the MUN domain and Syb2 cytoplasmic domain, Syb2 (29–96, S61C), Syb2$^{WD}$ (29–96, S61C/W89D/W90D), Syb2$^{RKAA}$ (29–96, S61C/R86A/K87A), and Syb2$^{4M}$ (29–96, S61C/R86A/K87A/W89D/W90D) were labeled with BODIPY FL N-(2-aminoethyl)-maleimide (BDPY) separately according to the manufacturer's instruction (Molecular Probes). Nonlinear curve fits were performed using the Hill equation, with the Hill coefficient fixed to 1. For the Syb2$^{49-96}$ displacement assays shown in Supplementary Fig. 2, BDPY-labeled Syb2$^{49-96}$ (S61C) was pre-assembled with Syx1 (residues 2–253) and SN25 (ΔN-complex (Syx1/SN25/Syb2$^{49-96}$)). Unlabeled Syb2 (residues 29–96, 2 µM) and its mutants were added to pre-assembled ΔN-complex (0.5 µM) to detect the displacement of Syb2$^{49-96}$. For the transition from the Munc18-1/Syx1$^{LE}$ complex to the SNARE complex in the absence of the MUN domain, co-expressed Munc18-1/Syx1$^{LE}$ complex (5 µM) was mixed with 10 µM Syb2 (residues 29–96) and 0.2 µM BDPY-labeled SN25. All of the experiments were performed at 30 °C.

**Liposome preparation**. 1-Palmitoyl-2-oleoyl-sn-glycero-3-phosphocholine (POPC), 1-palmitoyl-2-oleoyl-sn-glycero-3-phosphoethanolamine (POPE), 1,2-dioleoyl-sn-glycero-3-phospho-L-serine (sodium salt) (DOPS), 1,2-dipalmitoyl-sn-glycero-3-phosphoethanolamine-N-[lissamine rhodamine B sulfonyl] ammonium salt (rhodamine-PE), 1,2-dipalmitoyl-sn-glycero-3-phosphoethanolamine-N-[7-nitro-2-1,3-benzoxadiazol-4-yl] ammonium salt (NBD-PE), L-α-phosphatidylinositol-4,5-bisphosphate (Brain, Porcine) (ammonium salt) (PI[4,5]P2), cholesterol (ovine wool), 1,2-dipalmitoyl-sn-glycero-3-phosphoethanolamine-N-(cap biotinyl) (sodium salt) (Biotin-PE), and 1,2-dioleoyl-sn-glycerol (DAG) were all purchased from Avanti Polar Lipids. 1,1′-Dioctadecyl-3,3,3′,3′-tetramethylindocarbocyanine perchlorate (DiI) and 1,1′-dioctadecyl-3,3,3′,3′-tetramethylindodicarbocyanine perchlorate (DiD) were obtained from Molecular Probes.

For liposome fusion assays, donor liposomes (Syb2) contain 40% POPC, 17% POPE, 20% DOPS, 20% cholesterol, 1.5% NBD-PE, and 1.5% Rhodamine-PE. Acceptor liposomes (Munc18-1/Syx1) contain 55% POPC, 20% POPE, 18% DOPS, 2% PIP2, and 5% DAG. The lipid mixtures were dried with nitrogen flow in the glass tubes and then followed by vacuuming for 4 h. The lipid films were dissolved in 25 mM HEPES-K+, pH 7.4, 150 mM KCl, 10% glycerol (v/v), 0.2 mM TCEP, 1% CHAPS (Amresco) for 5 min, and then the purified Munc18-1/Syx1 full-length protein, Syb2 full-length protein or their mutations, and synaptotagmin-1 (Syt1) full-length protein were added into the mixture separately in a protein-to-lipid ratio of 1:500, 1:200, and 1:800, respectively. The proteoliposomes were incubated at room temperature for 40 min and then dialyzed in 25 mM HEPES-K+, pH 7.4, 150 mM KCl, 10% glycerol (v/v), 0.2 mM TCEP supplied with 1 g/L Bio-beads SM2 (BioRad) three times at 4 °C to remove the detergent.

For single-vesicle tethering assays, PM-vesicles contain 51% POPC, 19.5% POPE, 20% DOPS, 5% DAG, 2% PI(4,5)P₂, 0.5% Biotin-PE, and 2% DiD and SV-vesicles contain 63% POPC, 20% POPE, 15% DOPS, and 2% DiI. The lipid mixtures were dried with nitrogen flow in the glass tubes and then followed by vacuuming for 4 h. PM-vesicles were prepared through extrusion by a mini extruder (Avanti Polar Lipids) equipped with 200-nm-pore polycarbonate films

(Whatman). SV-vesicles were prepared using the same method in preparing liposomes for liposome fusion assays with a protein-to-lipid ratio of 1:200.

**Liposome fusion assay**. Donor liposomes (50 μM total lipids) were mixed with acceptor liposome (100 μM total lipids) in the presence of 5 μM SN25, 1 μM Munc13-1 $C_1$-$C_2$B-MUN fragment, and 1 mM $CaCl_2$. The experiments were performed at 37 °C and the NBD fluorescence was monitored with a PTI QM-40 fluorescence spectrophotometer at an excitation and emission wavelengths of 460 and 538 nm, respectively.

**HEK293T cell culture and lentiviruses preparation**. HEK293T cells (CRL-11268, ATCC) were cultured in a 37 °C incubator supplied with 5% $CO_2$, used for virus production. The medium contains Dulbecco's modified Eagle's medium (Gibco) and 10% fetal bovine serum (Gibco). Transfection of the lentiviral expression plasmid and three helper plasmids (pRSV-REV, pMDLg/pRRE, and pVSVG) was performed in six-well plates. The total DNA used for transfection was 4.5 μg/well by using the polyethylenimine (1 mg/ml in $ddH_2O$). Two days after transfection, cell supernatants were collected and subsequently pre-cleaned with a $1000 \times g$ centrifuge. Then, virus was concentrated by using sucrose-containing buffer as described before[58]. Briefly, the virus-containing supernatant was overlaid on a sucrose-containing buffer (50 mM Tris-Cl, pH 7.4, 100 mM NaCl, 0.5 mM EDTA) at a ratio of 4:1 (v/v). After centrifugation at 4 °C, the supernatants were carefully removed and the pellets were then dried. Phosphate-buffered saline was added for re-suspension in the 4 °C.

**Neuronal culture and virus infection**. The dissociated cortex neurons obtained from newborn pups of wild-type mice, digested by 0.25% trypsin-EDTA (Gibco) for 12 min at 37 °C, then were cultured on poly-ʟ-lysine-coated glass coverslips and maintained at 37 °C in 5% $CO_2$. The culture medium is minimum essential medium (Gibco) supplemented with 2% (v/v) B27 (Gibco), 0.5% (w/v) glucose (Sigma), 100 mg/l transferrin (Sigma), 5% (v/v) fetal bovine serum (Gibco), and 2 μM Ara-C (Sigma). Neurons were infected with lentivirus (described above) at days in vitro (DIV) 4–6 and electrophysiologically analyzed at DIV 13–14. All animal procedures were performed in accordance with South-Central University for Nationalities animal use rules and the requisite approvals of animal use committees.

**Electrophysiological recordings**. Electrophysiological recordings were performed as described elsewhere[21] using a HEKA EPC10 amplifier in whole-cell patch-clamp mode. Patch pipettes were prepared from borosilicate glass capillary tubes (World Precision Instruments, Inc.) by using a P-97 pipette puller (Sutter). The whole-cell pipette was filled with the solution, which contained 120 mM CsCl, 10 mM HEPES, 10 mM EGTA, 0.3 mM Na-GTP, and 3 mM $Mg^{2+}$-ATP (pH 7.2, adjusted with CsOH). The cell bath solution contained 140 mM NaCl, 5 mM KCl, 2 mM $MgCl_2$, 2 mM $CaCl_2$, 10 mM HEPES-NaOH, and 10 mM glucose (pH 7.4). The miniature IPSCs were monitored in the presence of tetrodotoxin (TTX, 1 μM) and 20 μM 6-cyano-7-nitroquinoxaline-2,3-dione (CNQX). The evoked IPSCs were recorded in the presence of 20 μM CNQX. Single extracellular stimulus pulses were given by 1 ms current injection (90 μA) with an Isolated Pulse Stimulator (Model 2100, A-M Systems Inc.). The RRP was measured by application of hypertonic sucrose (0.5 M). Synaptic responses were all monitored at a holding potential of −70 mV. The data were digitized at 10 kHz with a 2-kHz low-pass filter.

**Single-vesicle tethering assay**. General procedures were based on previous literature[35,36]. In brief, glass coverslips were coated with PEG and PEG-biotin (with a ratio of 9:1) and assembled into a flow chamber. Neutravidin (0.2 mg/ml; Pierce) were coated in each channel. The PM-vesicles were firstly immobilized on PEG/PEG-biotin surface during a 20-min incubation at 25 °C. After an extensive buffer wash step to remove the unbounded supported liposomes, 40 μM (total lipids) SV-vesicles and 1 μM Munc13-1 $C_1$-$C_2$B-MUN fragment were flowed into the channels and incubated for 30 min at 30 °C. Before imaging, unbounded SV-vesicles and proteins were washed out by a two-step buffer wash. Imaging was carried out with a Nikon Ti series inverted microscope equipped with TIRF illuminator (Nikon), beam splitter (Cairn Research), and an EMCCD camera (Andor iXon DU-897). Total internal reflection was achieved by a 1.49 NA ×100 oil-immersed objective (Nikon). PM-vesicle density was initially checked upon excitation at 640 nm. SV-vesicles were counted by the number of fluorescent spots from DiI dyes upon excitation at 532 nm. Images were processed by home-written MATLAB (Math-Works) script.

**Sequence alignment**. Sequence alignment was performed using Clustal Omega[59].

**Statistical analysis**. Prism 6.01 (GraphPad) and Image J (NIH) were used for graphing and statistical tests.

**Reporting summary**. Further information on experimental design is available in the Nature Research Reporting Summary linked to this article.

## Data availability

Data for the structure reported here have been deposited in the PDB under the accession number 6A30. A reporting summary for this Article is available as a Supplementary Information file. The source data underlying Figs. 1e, 1g, 2b, 2d, 3b, 4b, 4d, 5b, 5d–g, 6b, 6c, 7i, j and Supplementary Fig 9 are provided as a Source Data file. All other data supporting the findings of this study are available from the corresponding author on reasonable request.

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

## Acknowledgements
We thank the Shanghai Synchrotron Radiation Facility (SSRF) BL19U1 for helping with data collection. We thank Jiajie Diao for the technical help on single-vesicle experiments. We also thank Josep Rizo, Jizhong Lou, Sheng Wang, and Tao Xu for insightful discussions. This work was supported by the grants from the National Natural Science Foundation of China (31670846, 31721002, and 31670850), the National Key Basic Research Program of China (2015CB910800), and funds from Huazhong University of Science and Technology.

## Author contributions
S.W. and Y.L. generated all mutants and performed the in vitro binding and functional experiments. S.W., Y.L., S.Y., and R.Z. performed structural biology experiments. J.G. and X.Y. performed in vivo electrophysiology experiments. C.M. conceived the experiments. S.W. and C.M. wrote the manuscript.

## Additional information

**Competing interests:** The authors declare no competing interests.

