## [Peer Review File · Nature Communications]

Parts of this peer review file have been redacted as indicated due to copyright infringement.

Reviewers' comments:

Reviewer #1 (Remarks to the Author):

In this study, Wang et al use a powerful, multipronged approach to provide evidence that the MUN domain of Munc13 interacts with the linker region of synaptobrevin 2, identify the specific residues involved, and provide evidence that these interactions are required for formation of functional SNARE complexes. The experiments utilize a mutational approach to define regions of synaptobrevin and Munc13 important for the dissolution of the M18/syx complex. The choice of domains was heavily guided by existing structural data and also by new structural data presented by the authors. Results of functional assays are congruent with the biochemical results. The proposed working model of the final steps of SNARE complex formation provides exciting new details about late molecular transitions leading to the primed state. Overall, this study has the potential to advance the field in a significant way. On the down side, there are a few issues with the data that need to be addressed.

Specific Comments

1. Much of the evidence relies heavily on native-PAGE assays and the disassembly of the M18/syx complex. To better evaluate/interpret these data, the authors should identify the relevant bands by western blot. In addition, the authors should comment on the numerous other bands that are not identified. What are they and what, if any, is their significance? Are these degradation products, non-productive interactions between proteins or non-specific aggregates? For example, what is the band in Figures 1E and 2A that runs above the M18/Syx band observed in all samples? Also, is there a band that corresponds to the SNARE complex that can be observed and quantified?

2. Integrated density figures. With the new push for rigor and reproducibility, technical replicates on the same sample is no longer considered sufficient evidence of an effect. One must perform the experiment multiple times. After that has been done, the statistical analysis on the data must be performed and the data presented in a way that indicates not just the mean or median, but also the range of data.

3. Throughout the manuscript, the authors assume that if the M18/syx-1 complex is no longer present, a SNARE complex must have formed. While this is a reasonable expectation given our models of how this should work, corroborative evidence that SNARE complexes do form would strengthen this assertion.

4. Controls: A good control for the data shown in Figure 5 would be to test the mutations that did not lead to dissociation of the M18/syx complex in parts A and B and show that they also fail to catalyze fusion. In the supplemental Figure 2, one could also include data to establish that in the authors hands neither Syb2N nor Syb2C alone are capable of forming a SNARE complex. These data would serve as a functional control for the data shown in Figure 2C.

5. The physiological data shown in Figure 5 convincingly show that D1358K gives a smaller releasable pool. Ultrastructural data would further distinguish between a paucity of vesicles located at the active zone (e.g. docking defect) versus a defect exclusively in priming. Presumably, without Mun, the vesicles may not even dock, as suggested by the data in Figure 7h. Thus, the authors need to be very careful when concluding where the actual defect lies (docking versus priming) based on the physiological data. It could be a docking defect that leads to a priming defect.

Minor Comments

1. The manuscript has a number of grammatical issues, the corrections to which should be performed carefully by a scientist who is a native english speaker in order to preserve scientific meaning.

2. It would be very helpful to have a graphic depiction or a table that summarizes the different mutants that have been examined included within the main manuscript.

Reviewer #2 (Remarks to the Author):

In this study, the authors address several issues related to Munc13 and Munc18 function in SNARE complex formation. In any assay that detects M18/Syx dissociation which the authors interpret as Syx transition into a SNARE complex, they find a requirement for N-terminal Syb2 and C-terminal SN25 sequences. This is interpreted as a requirement for half zippering of SNARE complexes for Syx transitions. In additional studies, they identify a linker region of Syb that is critical for transition. In the core study of the paper, a crystal structure of MUN with segment D incorporating a Syb linker protein is shown. The functional significance of MUN segment D residues and linker residues in Syb is explored in liposome fusion, synaptic transmission, and liposome binding are explored. The

manuscript provides several important new findings for understanding accessory factor function in SNARE-dependent fusion. Additional clarification is needed on a number of points as summarized below. In general, figure legends were sparse, which will impede efforts to reproduce the work.

1. The authors rely on an assay (shown in Fig 1c) where MUN addition dissociates a M18/Syx complex. However, it appears that this assay relies upon truncated soluble SNARE proteins. It is known that M18 exhibits a very high binding affinity to truncated soluble Syx. However, M18 also binds full length Syx on membrane but at orders of magnitude lower affinity. Hence, one could question the relevance of the assay performed as in Fig 1c. Have the authors performed similar studies on M18/full length Syx in membranes to detect an effect of MUN on dissociation?

The authors interpret their assay on M18/Syx dissociation by MUN as leading to SNARE complex assembly (Fig 1b and text). However, SNARE complex assembly is not shown in any of the native PAGE studies. Thus, in panel 1c, there is no shift in SN25 into a complex, Syb2 wt is difficult to see at all, and Syx disappears from M18/Syx but it is not clear where it goes. So it is not evident where a SNARE complex is on this gel system. The authors need to further document where SNARE complexes are on these native gels or alternatively independently assess SNARE complex formation induced by MUN under the various conditions in Figs 1c, 1e, 2a, 2c, 4a, 5a, 6a so that they can use the word “transition” as opposed to M18/Syx dissociation.

While it may have been shown in prior publications, the authors need to show that the M18/Syx dissociation assay depends on both SN25 and Syb2 in this paper because interpretations depend on it. It would also be important to show MUN stimulation of SNARE complex formation starting with either Syx or M18/Syx. An alternative interpretation of this assay, if there is MUN stimulation of SNARE complex formation, is that MUN enhances SNARE complex formation to drive Syx out of a M18/Syx complex. Overall, the question here is whether Munc13-1 has a major M18-independent role.

2. It is not clear that the data in Supplementary Fig 1 are relevant to the studies of Fig 1. This displacement/anisotropy assays may measure properties of Syb2 but there is no MUN or M18 in these studies. It is based on spontaneous rather than chaperoned SNARE complex assembly.

3. Fig 1: the C-terminal helix of SNAP25 is usually termed SN2 rather than SN3 (?).

4. I. 144. “Considering that ... the strong defect on the transition might arise likely because they impair the Munc18-1 interaction ...” The meaning of this sentence is unclear.

3. Many of the Fig legends incompletely describe the studies. As only one example of many examples, Fig S3 and Fig S4 does not indicate what fraction of input MUN was run on gel. The concentration of MUN used in these experiments is also not indicated. While Fig 3b shows an estimate of 13 micromolar Kd for MUN binding to Syb2, there is no indication of the stoichiometry of this interaction. Thus, in Fig 3a it is likely that a very small fraction of MUN was bound to GST-Syb2 at this Kd even if very high concentrations of MUN were used by the time a GST pulldown n was washed. The reader should be given some sense of this by providing the fraction of input and bound run on the gel.

4. Fig 3c,d appears to be the highlight of this article directly showing the MUN crystal with Syb2 peptide integrated into it. Densities for W89/W90 but not R86/K87 were observed. Given the data in Fig 3a and Fig 4, one wonders whether a mutant Syb2 peptide was used as a negative control in these studies. Were any other Syb2 peptides used as a control in these studies?

5. It would importantbe quite interesting to know whether Munc13-1 dissociates from SNARE complexes following its actions. The authors attempt to address this issue indirectly in Fig S4 where they show that MUN fails to bind SNARE complexes but does bind Syb2. This is not a very convincing way to address this question especially since other investigators have previously reported MUN binding to SNARE complexes.

Reviewer #3 (Remarks to the Author):

This manuscript by Wang et al., presents more important clues in our understanding of how the SNARE-regulatory proteins Munc13 and Munc18 collaborate to drive proper neuronal SNARE complex assembly. This area of study has been confusing and contradictory for many years, partly due to the complicated machinery with numerous disparate binding sites. The data presented in this manuscript indicate that the interaction between the MUN domain of Munc13 with the C-terminal linker of synaptobrevin is critical for the transition of the Munc18-closed Syx complex to zippered SNARE complex, a key intermediate in specific SNARE complex assembly. This conclusion is well supported by a new crystal structure, mutagenesis, biochemical binding, tethering, and fusion experiments, and by in vivo electrophysiology measurements of fusion activity.

Mostly minor comments:

1) p. 5, line 97: a better description of the terminology of the truncation mutations with regard to the layers would be helpful, and make sure that they are being used consistently throughout the text, figure and legend. Similarly, the Syb2 -7 mutant (L32A) was not described.

2) in addition, all the figure legends could use additional information/details to make them understandable without having to constantly refer back to the text.

3) p. 5, line 100: although the native-PAGE assay was previously published, it only appeared in the supplemental materials, and could use a little extra explanation here. Also, it would be useful to label the other bands in the gels (if known). In the quantification (e.g. 1d and all others) graphs, it is unclear what exactly is plotted on the y-axis.

4) p. 6, line 109: a better description of the displacement assay would be useful

5) p. 9 and 13: the L70 and F77 residues of Syb2, apparently are the Munc18 binding site, and mutants block Munc18 binding, but this data is not in ref 30. As this is important information for understanding how Munc18 and Munc13 may or may not interact together on Syb2, this needs to be clarified. This is especially true to help understand Fig 6C (why is Munc18/Syx binding to Syb2 so poor?).

6) p. 9, line 170 and Fig 3b: the K_d for the MUN-Syb2 binding is confusing. How was the data fitted to calculate that K_d ? It seems to have more than one binding site? And are the lab's pipets really that accurate (to those significant digits)?

7) What is the binding constant for the shorter peptide that was used for the crystal structure? Does the peptide contain the entire binding site or just part? It doesn't look like enough interaction to give 12 μ M binding affinity. Can the missing amino acids (or even a longer fragment of Syb2) be modeled into the structure? Is there evidence that Syb becomes helical (or not) when bound to MUN? This information might help form the model in Suppl Fig 6. This is an important panel and would be better as part of Fig 8 in the main paper.

8) most of the manuscript could use a little grammatical copy editing, but otherwise is well written and clear

Reviewers' comments:

Reviewer #1 (Remarks to the Author):

In this study, Wang et al use a powerful, multipronged approach to provide evidence that the MUN domain of Munc13 interacts with the linker region of synaptobrevin 2, identify the specific residues involved, and provide evidence that these interactions are required for formation of functional SNARE complexes. The experiments utilize a mutational approach to define regions of synaptobrevin and Munc13 important for the dissolution of the M18/syx complex. The choice of domains was heavily guided by existing structural data and also by new structural data presented by the authors. Results of functional assays are congruent with the biochemical results. The proposed working model of the final steps of SNARE complex formation provides exciting new details about late molecular transitions leading to the primed state. Overall, this study has the potential to advance the field in a significant way. On the down side, there are a few issues with the data that need to be addressed.

We thank the reviewer #1 for the summary of the paper and for the praise of our work. The point -to-point replies are as follow:

Specific Comments

1. Much of the evidence relies heavily on native-PAGE assays and the disassembly of the M18/syx complex. To better evaluate/interpret these data, the authors should identify the relevant bands by western blot. In addition, the authors should comment on the numerous other bands that are not identified. What are they and what, if any, is their significance? Are these degradation products, non-productive interactions between proteins or non-specific aggregates? For example, what is the band in Figures 1E and 2A that runs above the M18/Syx band observed in all samples? Also, is there a band that corresponds to the SNARE complex that can be observed and quantified?

We have added a new figure to interpret the bands (see below, or Figure. S1 in the revised manuscript). To better clarifying our point, every single protein and possible combination of the components were load ed and referenced in our new native-PAGE shown below. For the reviewer's specific concerns, the band that runs above the

M18-1/Syx1 band should be the non-productive M18-1/Syx1 aggregates (lane 7); the MUN domain, Munc18-1 or free synaptobrevin-2 (Syb2) exhibits smear-type bands (lane 2, 4 and 5); free syntaxin-1 (Syx1) shows multi-bands because of different assembly/aggregation states (lane 1). We could see the sharp SNARE complex bands when we mixed the three SNAREs merely (lane 6). However, in the presence of Munc18 and the MUN domain, the SNARE complex displays a smear-type band that runs slower than the Munc18/MUN-free SNARE complex (lane 14 vs. lane 6). This might due to a cooperative interaction among the SNARE complex, Munc18 and the MUN domain, as previously suggested (*Ma et al., Nat. Struct. Mol. Biol. 2011*). Additionally, it is notable that a variety of SNARE truncations (for SN25 and Syb2) were used in this work. These long and short truncations are predicted to form multiple SNARE complexes with different sizes and charges, and lead to ambiguous bands on native-PAGE that are difficult to quantify. These are the reasons why we evaluate our data with the M18 -1/Syx1 complex rather than the SNARE complex. We interpret all of these in the new Figure S1 legend.

Figure S1. Standard example of the native-PAGE assay. Asterisk indicates a putative non-productive aggregation of Munc18-1/Syx1 complex. The SNARE complex in last lane displays a smear -type band compared to the sixth lane.

2. *Integrated density figures. With the new push for rigor and reproducibility, technical replicates on the same sample is no longer considered sufficient evidence of an effect. One must perform the experiment multiple times. After that has been done, the statistical analysis on the data must be performed and the data presented in a way that indicates not just the mean or median, but also the range of data.*

We feel so sorry to make it unclear. Actually we used the recombinant proteins to perform the experiment for at least three times, INSTEAD OF running PAGE three times with the same reaction sample or analyzing the same gel for three times. The quantifications were indeed done by assessing three individual performed gels. Anyway, we have removed the misleading word "technical replicates" and performed statistical analysis as well.

3. Throughout the manuscript, the authors assume that if the M18/syx-1 complex is no longer present, a SNARE complex must have formed. While this is a reasonable expectation given our models of how this should work, corroborative evidence that SNARE complexes do form would strengthen this assertion.

As shown in lane 14 of the above Figure S1, the disappearance of the M18 -1/Syx1 complex is accompanied by the appearance of the SNARE complex (a smear-type). In line with this point, the intensity of the M18-1/Syx1 complex band does not change at all if lacking any of the component (lanes 8–13); disappearance of the M18-1/Syx1 complex can only be detected when all the components are included (lane 14). These evidence indicate that the formation of the SNARE complex corresponds to the disassociation of the M18 -1/Syx1 complex. These data also show that the transition from the M18-1/Syx1 complex to the SNARE complex requires not only Munc13 catalysis but also the presence of SN25 and Syb2, which has been suggested in our previous work (Wang *et al.*, *EMBO J*, 2017). We added this point in the new Figure S1 legend.

4. Controls: A good control for the data shown in Figure 5 would be to test the mutations that did not lead to dissociation of the M18/syx complex in parts A and B and show that they also fail to catalyze fusion. In the supplemental Figure 2, one could also include data to establish that in the author's hands neither Syb2^N nor Syb2^C alone are capable of forming a SNARE complex. These data would serve as a functional control for the data shown in Figure 2C.

The D1358K mutation strongly reduced the activity of the MUN domain in catalyzing the transition from the M18/Syx1 complex to the SNARE complex (Fig. 5a, b). In line with this data, the same mutation strongly impaired the ability of C₁-C₂B-MUN in catalyzing liposome fusion (Fig. 5 c, d) and exocytosis (Fig. 5 e –g). These experiments were well done. As suggested, the controls that neither Syb2^N nor Syb2^C is capable of forming a SNARE complex have been included in Fig. S3 in the revised paper (corresponding to Fig. S2 in the initial version).

5. The physiological data shown in Figure 5 convincingly show that D1358K gives a smaller releasable pool. Ultrastructural data would further distinguish between a paucity of vesicles located at the active zone (e.g. docking defect) versus a defect exclusively in priming. Presumably, without Mun, the vesicles may not even dock, as suggested by the data in Figure 7h. Thus, the authors need to be very careful when concluding where the actual defect lies (docking versus priming) based on the physiological data. It could be a docking defect that leads to a priming defect.

This is a really interesting question. We have to admit that it is technically hard to distinguish docking with priming step by physiological experiments, as a defect on either docking or priming leads to a strongly reduced RRP. And this would hinder us to pinpoint at which stage of exocytosis (docking or priming) the MUN –Syb2 interaction begins to be engaged. In addition to the essential role of this interaction in SNARE complex formation (the priming step) illustrated in most of our data, the data in Fig. 7 that the MUN–Syb2 interaction has an ability to bring the two membranes into close proximity raise a possibility that this interaction might be involved in vesicle docking. Since distinguishing docking and priming defect is beyond the scope of the present manuscript, we'd like to change the word “priming” to “docking/priming” throughout the manuscript.

Minor Comments

1. *The manuscript has a number of grammatical issues, the corrections to which should be performed carefully by a scientist who is a native English speaker in order to preserve scientific meaning.*

We have carefully checked the grammar with the help of some English speakers, and the writing has been improved by *Nature Research Editing Service*.

2. *It would be very helpful to have a graphic depiction or a table that summarizes the different mutants that have been examined included within the main manuscript.*

We thank the reviewer for this advice. We have added a table (Table S1) to summarize all the mutants used in our experiments in the revised manuscript.

Reviewer #2 (Remarks to the Author):

In this study, the authors address several issues related to Munc13 and Munc18 function in SNARE complex formation. In any assay that detects M18/Syx dissociation which the authors interpret as Syx transition into a SNARE complex, they find a requirement for N-terminal Syb2 and C-terminal SN25 sequences. This is interpreted as a requirement for half zippering of SNARE complexes for Syx transitions. In additional studies, they identify a linker region of Syb that is critical for transition. In the core study of the paper, a crystal structure of MUN with segment D incorporating a Syb linker protein is shown. The functional significance of MUN segment D residues and linker residues in Syb is explored in liposome fusion, synaptic transmission, and liposome binding are explored. The manuscript provides several important new findings for understanding accessory factor function in SNARE - dependent fusion. Additional clarification is needed on a number of points as summarized below. In general, figure legends were sparse, which will impede efforts to reproduce the work.

We thank the reviewer #2 for the summary of the paper and for the praise of our work. The point -to-point replies are as follow:

1. *The authors rely on an assay (shown in Fig 1c) where MUN addition dissociates a M18/Syx complex. However, it appears that this assay relies upon truncated soluble SNARE proteins. It is known that M18 exhibits a very high binding affinity to truncated soluble Syx. However, M18 also binds full length Syx on membrane but at orders of magnitude lower affinity. Hence, one could question the relevance of the assay performed as in Fig 1c. Have the authors performed similar studies on M18/full length Syx in membranes to detect an effect of MUN on dissociation?*

In our experimental conditions, the transition from the M18-1/Syx1 complex to the SNARE complex is strictly dependent on Munc13-1 catalysis, regardless of using the cytosolic domain of Syx1 or full-length Syx1. As shown in Fig. 4C and Fig. 5C, when deletion of C₁-C₂B-MUN, no lipid mixing was observed between liposomes reconstituted with M18-1/Syx1 (full-length) complex and liposomes containing Syb2 in the presence of SN25 and Syt1/Ca²⁺, suggesting an essential role of the MUN domain in opening of the M18-1/Syx1 (full-length) complex on the membrane. In our previous work, we also observed that, when starting with the M18 -1/Syx1 (full-length) complex embedded on the membrane, C₁-C₂B-MUN still play an important role in promoting the transition to the SNARE complex in the presence of SN25 and Syb2 (29–96) (*Li et al., Front. mol. neurosci., 2017, Figure 2*).

The authors interpret their assay on M18/Syx dissociation by MUN as leading to SNARE complex assembly (Fig 1b and text). However, SNARE complex assembly is not shown in any of the native PAGE studies. Thus, in panel 1c, there is no shift in SN25 into a complex, Syb2 wt is difficult to see at all, and Syx disappears from M18/Syx but it is not clear where it goes. So it is not evident where a SNARE complex is on this gel system. The authors need to further document where SNARE complexes are on these native gels or alternatively independently assess SNARE complex formation induced by MUN under the various conditions in Figs 1c, 1e, 2a, 2c, 4a, 5a, 6a so that they can use the word “transition” as opposed to M18/Syx dissociation.

We need to apologize that we didn't put enough controls in our native-PAGE assays. Now, we provide a new figure to interpret the bands (Figure S1 in the revised manuscript). The amount of SN25 used in all of our native -PAGE assays is 10 μ M, which is 5-fold excess to the amount of the M18-1/Syx1 complex (2 μ M) (more details please see Materials and Methods). This explains why most SN25 does not shift into the SNARE complex in the gel. In addition, Syx1 alone displays multi-bands (lane 1), and Syb2 displays a smear band that is hard to detect (lane 4). We could see the sharp SNARE complex bands when we mixed the three SNAREs merely (lane 6). However, in the presence of Munc18 and the MUN domain, the SNARE complex displays a smear -type band that runs slower than the Munc18/MUN-free SNARE complex (lane 14 vs. lane 6). This might due to a cooperative interaction among the SNARE complex, Munc18 and the MUN domain, as previously suggested (*Ma et al., Nat. Struct. Mol. Biol. 2011*). Additionally, it is notable that a variety of SNARE truncations (for SN25 and Syb2) are used in this work. These long and short truncations are predicted to form multiple truncated SNARE complexes with different sizes and charges, and lead no doubt to ambiguous bands on native-PAGE that are difficult to quantify. These are the reasons why we evaluate our data with the band of the M18-1/Syx1 complex rather than that of the SNARE complex.

Figure S1. Standard example of the native-PAGE assay. Asterisk indicates a putative non-productive aggregation of Munc18-1/Syx1 complex. The SNARE complex in last lane displays a smear-type band compared to the sixth lane.

As shown in lane 14 of the above Figure S1, the disappearance of the M18 -1/Syx1 complex is accompanied by the appearance of the SNARE complex (a smear-type). In line with this point, the intensity of the M18-1/Syx1 complex band does not change at all if lacking any of the component (lanes 8 –13). Disappearance of the M18-1/Syx1 complex can only be detected when all the components are included (lane 14). These evidence indicate th at the formation of the SNARE complex corresponds to the disassociation of the M18 -1/Syx1 complex. These data also show that the transition from the M18-1/Syx1 complex to the SNARE complex requires not only Munc13 catalysis but also the presence of SN25 and Syb2, as suggested in our previous work (Wang *et al.*, *EMBO J*, 2017).

While it may have been shown in prior publications, the authors need to show that the M18/Syx dissociation assay depends on both SN25 and Syb2 in this paper because interpretations depend on it. It would also be important to show MUN stimulation of SNARE complex formation starting with either Syx or M18/Syx. An alternative interpretation of this assay, if there is MUN stimulation of SNARE complex formation, is that MUN enhances SNARE complex formation to drive Syx out of a M18/Syx complex. Overall, the question here is whether Munc13-1 has a major M18-independent role.

We have shown the requirement of both SN25 and Syb2 in the transition of the M18 -1/Syx1 complex to the SNARE complex (see Figure S1 and the above discussion), and this notion actually has been suggested in our previous work (Wang *et al.*, *EMBO J*, 2017). We have noted this point in the Introduction part and in the main text (e.g. lines 55–57; 84–86; 101–105). In addition, we have previously revealed that the MUN domain fails to accelerate SNARE complex assembly when starting with isolated Syx1 in our previous work (Ma *et al.*, *Nat. Struc. Mol. Biol.*, 2011, supplementary Figure 9f).

[Redacted]

2. *It is not clear that the data in Supplementary Fig 1 are relevant to the studies of Fig 1. This displacement/anisotropy assays may measure properties of Syb2 but there is no MUN or M18 in these studies. It is based on spontaneous rather than chaperoned SNARE complex assembly.*

First, it was proposed that SNARE complex assembly initiates with N-terminal nucleation and proceeds in an N- to C-terminal propagation manner. A key *in-vitro* experiment to support this notion is the Syb2^{49–96} displacement assay, which was established by Dirk Fasshauer's lab (Pobbati *et al.*, *Science*, 2006). In this displacement assay, the preassembled ΔN-complex (Syx1/SN25/Syb2^{49–96}), with its N-terminal SNARE binding layers being exposed, is accessible for N-terminal nucleation and zippering with Syb2 (29–96); and subsequent N- to C-zipper of Syb2 (29–96) leads eventually to the disassociation of Syb2^{49–96} from the ΔN-complex. As a consequence, the N-terminal nucleation/zippering ability of Syb2 (29–96) can be easily characterized by assessing the disassociation rate of

fluorescence-labeled Syb2⁴⁹⁻⁹⁶ with a time-based fluorescence anisotropy.

[Redacted]

Second, we'd like to mention that increasing evidence suggest a functional relevance between Syb2⁴⁹⁻⁹⁶ and M18/MUN in "chaperoned" SNARE complex assembly. (I) based on the closed M18-1/Syx1 complex structure, the C-terminal SNARE layers of Syx1 are intensively clamped by M18-1; (II) single-molecule FRET results in our recent study (Wang *et al.*, *EMBO J*, 2017) showed that the MUN domain does not dissociate the C-terminal SNARE layers of Syx1 from M18-1 clamping, but rather induces a conformational change in the linker region and the following N-terminal SNARE layers of Syx1 (see the Figure above, from Wang *et al.*, *EMBO J*, 2017). Thus, in Syb2⁴⁹⁻⁹⁶-guided (Syx1/SN25/Syb2⁴⁹⁻⁹⁶) and M18-M13-guided SNARE assemblies, both Syb2⁴⁹⁻⁹⁶ and M18/M13 would ensure Syb2 (29-96) zippering to begin strictly at the N-terminal SNARE layers, allowing us to characterize SNARE N-terminal nucleation and zippering via introducing multiple mutations. Thus, both Syb2⁴⁹⁻⁹⁶-guided (Syx1/SN25/Syb2⁴⁹⁻⁹⁶) and M18-M13-guided SNARE assemblies should be regarded as "chaperoned" SNARE complex assembly. Instead, for "spontaneous" SNARE complex assembly, it only contains Syx1, SN25 and Syb2 (29-96) (without Syb2⁴⁹⁻⁹⁶ or M18/M13). Indeed, the requirement of layers -7 and -6 of Syb2 (29-96) for driving

49-96

SNARE N-terminal nucleation and zippering, which was detected by the Syb2

displacement assay, can be

reflected in M18-M13-guided SNARE complex assembly (Fig 1c, d), but not in "spontaneous" SNARE complex assembly (see the Figure below). (Note that in spontaneous SNARE complex assembly, all binding layers of the SNARE motifs are exposed in solution, so the three SNAREs should have a strong capacity to assemble even without the layer -7 and -6).

3. Fig 1: the C-terminal helix of SNAP25 is usually termed SN2 rather than SN3 (?).

We have changed all “SN3” with “SN2” throughout the manuscript.

4. l. 144. “Considering that ... the strong defect on the transition might arise likely because they imp air the Munc18-1 interaction ...” The meaning of this sentence is unclear.

We deleted this sentence in the revised manuscript.

3. Many of the Fig legends incompletely describe the studies. As only one example of many examples, Fig S3 and Fig S4 does not indicate what fraction of input MUN was run on gel. The concentration of MUN used in these experiments is also not indicated. While Fig 3b shows an estimate of 13 micromolar Kd for MUN binding to Syb2, there is no indication of the stoichiometry of this interaction. Thus, in Fig 3a it is likely that a very small fraction of MUN was bound to GST-Syb2 at this Kd even if very high concentrations of MUN were used by the time a GST pulldown n was washed. The reader should be given some sense of this by providing the fraction of input and bound run on the gel.

We thank the reviewer's advices. Now all figure legends are detailed and improved. For Fig. 3b, we used Hill equation to achieve non-linear curve fit. As revealed by the crystal structure, the stoichiometry of the interaction is 1:1, so we fixed the Hill coefficient to 1 ($n = 1$). We have described it in the revised manuscript. In addition, the fraction of input are added in Fig. 3a.

4. Fig 3c,d appears to be the highlight of this article directly showing the MUN crystal with Syb2 peptide integrated into it. Densities for W89/W90 but not R86/K87 were observed. Given the data in Fig 3a and Fig 4, one wonders whether a mutant Syb2 peptide was used as a negative control in these studies. Were any other Syb2 peptides used as a control in these studies?

Syb2 peptide (residues 75–93) was only used in our crystallization experiments (Fig 3c, d). In Fig. 3a, b and Fig. 4a, b, the negative control are Syb2^{RKAA} (residues 29–96, R86A/K87A), Syb2^{RKAA} (residues 29–96, W89D/W90D) and Syb2^{4W} (residues 29–96, R86A/K87A/W89D/W90D). We made it more clearly in the revised manuscript.

5. It would importantbe quite interesting to know whether Munc13-1 dissociates from SNARE complexes following its actions. The authors attempt to address this issue indirectly in Fig S4 where they show that MUN fails to bind SNARE complexes but does bind Syb2. This is not a very convincing way to address this question especially since other investigators have previously reported MUN binding to SNARE complexes.

Now we stated in our manuscript (lines 169–172): “This binding result suggests that the MUN domain might disassociate from the LR of Syb2 upon the final zippering of the SNARE complex into the membrane, regardless of the likely maintained interaction between the MUN domain and the four -helix SNARE bundle, as detected by NMR experiment.”, we do not conclude that the MUN domain disassociates from the SNARE complex, but suggest that it may disassociates from the linker region of Syb2. Actually, by using NMR 2D -HSQC we previously showed that there exists a cooperative interaction between the SNARE complex, M18 and MUN, where MUN prefers to bind the SNARE four helical bundle (Ma et al., Nat. Struc. Mol. Biol., 2011). This interaction is pretty weak without M18, and is hard to be detected in GST pull-down experiment. We believe that it is an open question in this field and we'd like to address it in the future.

Reviewer #3 (Remarks to the Author):

This manuscript by Wang et al., presents more important clues in our understanding of how the SNARE-regulatory proteins Munc13 and Munc18 collaborate to drive proper neuronal SNARE complex assembly. This area of study has been confusing and contradictory for many years, partly due to the complicated machinery with numerous disparate binding sites. The data presented in this manuscript indicate that the interaction between the MUN domain of Munc13 with the C-terminal linker of synaptobrevin is critical for the transition of the Munc18 -closed Syx complex to zippered SNARE complex, a key intermediate in specific SNARE complex assembly. This conclusion is well supported by a new crystal structure, mutagenesis, biochemical binding, tethering, and fusion experiments, and by in vivo electrophysiology measurements of fusion activity.

We thank the reviewer #3 for the summary of the paper and for the praise of our work. The point -to-point replies are as follow:

Mostly minor comments:

1) p. 5, line 97: a better description of the terminology of the truncation mutations with regard to the layers would be helpful, and make sure that they are being used consistently throughout the text, figure and legend. Similarly, the Syb2 - 7 mutant (L32A) was not described.

Now we have added a table (Table S1) to summarize all the mutants used in our experiments.

2) in addition, all the figure legends could use additional information/details to make them understandable without having to constantly refer back to the text.

As suggested, all the figure legends have been detailed and improved.

3) p. 5, line 100: although the native-PAGE assay was previously published, it only appeared in the supplemental materials, and could use a little extra explanation here. Also, it would be useful to label the other bands in the gels (if known). In the quantification (e.g. Id and all others) graphs, it is unclear what exactly is plotted on the y-axis.

We have added a new figure (Fig. S1, see below) to illustrate every single protein or protein combination in our native-PAGE assay, and described this assay more clearly. The amount of SN25 used in all of our native-PAGE assays is 10 μ M, which is 5-fold excess to the amount of the M18-1/Syx1 complex (2 μ M) (more details please see Materials and Methods). This explains why most SN25 does not shift into the SNARE complex in the gel. In addition, Syx1 alone displays multi-bands (lane 1), and Syb2 displays a smear band that is hard to detect (lane 4). We could see the sharp SNARE complex bands when we mixed the three SNAREs merely (lane 6). However, in the presence of Munc18 and the MUN domain, the SNARE complex displays a smear -type band that runs slower than the Munc18/MUN-free SNARE complex (lane 14 vs. lane 6).

In addition, the y-axis of all quantification graphs represents the normalized integrated gray level of each assessed M18-1/Syx1 band, which has been indicated in the revised figure legends.

Figure S1: Standard example of the native-PAGE assay. Asterisk indicates a putative non-productive aggregation of Munc18-1/Syx1 complex. The SNARE complex in last lane displays a smear-type band compared to the sixth lane.

4) p. 6, line 109: a better description of the displacement assay would be useful

As suggested, we add a detailed description in the main text and the figure legend.

5) p. 9 and 13: the L70 and F77 residues of Syb2, apparently are the Munc18 binding site, and mutants block Munc18 binding, but this data is not in ref 30. As this is important information for understanding how Munc18 and Munc13 may or may not interact together on Syb2, this need to be clarified. This is especially true to help understand Fig 6C (why is Munc18/Syx binding to Syb2 so poor?).

We added a sequence alignment (see below, and Fig. S4 in the revised manuscript), the L70 and F77 residues of Syb2 correspond to L194 and F201 of Nyv1. In the Vps33/Nyv1 structure (Baker *et al.*, *Science*, 2015), L194 and F201 of Nyv1 mediate Vps33 interaction.

```

G0S5G3 | Ct_Nyv1  181  GERIDLVDRKTRDRGGSSAREFRLRSRG LKRKMWWKNVKLMLGLGFFVVLIDLTETVTSVKG
P63045 | Rn_Syb2  57  DQKLISELDRAADANQAGASQFETSAAK LKRKYWWKNLKMMLTGLGVLICAITLITIVVFST

```

6) p. 9, line 170 and Fig 3b: the K_d for the MUN-Syb2 binding is confusing. How was the data fitted to calculate that K_d ? It seems to have more than one binding site? And are the lab's pipets really that accurate (to those significant digits)?

In Fig. 3b, we used Hill equation to achieve non-linear curve fit. As shown in the crystal structure, the stoichiometry of the interaction is 1:1, so we fixed the Hill coefficient to 1 ($n = 1$). Now the description has been

added to the figure legend. We feel sorry to over-interpret our data, now the value is fixed to $12.9 \pm 1.8 \mu\text{M}$.

7) *What is the binding constant for the shorter peptide that was used for the crystal structure? Does the peptide contain the entire binding site or just part? It doesn't look like enough interaction to give 12 μM binding affinity. Can the missing amino acids (or even a longer fragment of Syb2) be modeled into the structure? Is there evidence that Syb becomes helical (or not) when bound to MUN? This information might help form the model in Suppl Fig 6. This is an important panel and would be better as part of Fig 8 in the main paper.*

In Fig. 3b, Syb2 (29–96) binds MUN with $12.9 \pm 1.8 \mu\text{M}$ binding affinity. Although we didn't measure the binding constant between the Syb2 peptide (residues 75–93) and MUN, our biochemical and structural data provide enough evidence that residues 86–92 are responsible for MUN interaction: Fig. 3a showed that F77, K83 and K85 of Syb2 (29–96) do not involved in MUN interaction; Fig. 3c–f revealed that the residues R86, K87, Y88, W89, W90, K91, N92 are involved in MUN interaction. These lead us to suspect that there should be no missing amino acids in our described MUN-Syb2 interaction. Based on the crystal structure, MUN-bound Syb2 segment (residues 87–92) adopts a folded but not α -helical conformation.

8) *most of the manuscript could use a little grammatical copy editing, but otherwise is well written and clear*

We have improved the writing with help of editing company (*Nature Research Editing Service*).

Reference

Baker, R. W., Jeffrey, P. D., Zick, M., Phillips, B. P., Wickner, W. T., & Hughson, F. M. (2015). A direct role for the Sec1/Munc18-family protein Vps33 as a template for SNARE assembly. *Science*, *349*(6252), 1111-1114.

Li, Y., Wang, S., Li, T., Zhu, L., Xu, Y., & Ma, C. (2017). A stimulation function of synaptotagmin -1 in ternary SNARE complex formation dependent on Munc18 and Munc13. *Frontiers in molecular neuroscience*, *10*, 256.

Ma, C., Li, W., Xu, Y., & Rizo, J. (2011). Munc13 mediates the transition from the closed syntaxin–Munc18 complex to the SNARE complex. *Nature structural & molecular biology*, *18*(5), 542.

Pobbati, A. V., Stein, A., & Fasshauer, D. (2006). N-to C-terminal SNARE complex assembly promotes rapid membrane fusion. *Science*, *313*(5787), 673-676.

Wang, S., Choi, U. B., Gong, J., Yang, X., Li, Y., Wang, A. L., ... & Ma, C. (2017). Conformational change of syntaxin linker region induced by Munc13s initiates SNARE complex formation in synaptic exocytosis. *The EMBO journal*, e201695775.

We hope that these revisions are satisfactory and address the reviews' concerns. We would like to thank again the reviewers for their many useful comments, which have helped to substantially improve the manuscript. Thank you very much for handling the manuscript.

Sincerely,
Cong Ma
Huazhong University of Science and Technology

Reviewers' comments:

Reviewer #1 (Remarks to the Author):

In this revised manuscript, the authors have adequately addressed my previous concerns, and the revisions have raised no new concerns. The manuscript text revisions, from the grammatical changes to the expansion of the figure legends, increase readability. The new supplemental data and table provide important support for the conclusions contained within the main text.

Reviewer #2 (Remarks to the Author):

This manuscript has been improved by revision but there remains an important major concern (1) as well as a new minor concern (2).

1. Suppl Fig 1. The first part of this manuscript relies extensively on the assay shown in Fig. 1b that indicates a dissociation of a M18/Syx complex by the MUN domain. The dissociation required addition of Snb and SN25, and the authors term this dissociation a "transition" to the formation of Syx-containing SNARE complexes. However, there are other interpretations of these data. It was pointed out that SNARE complex formation was not evident in this native PAGE system. The authors responded by including Supp Fig 1 that annotates various protein bands in these reactions. Unfortunately these data are unclear because of the appearance of smeared bands. Of concern is the fact that the heavily-stained M18/Syx bands disappear and do not appear elsewhere and there is no corresponding increase that could represent the SNARE complexes. A very simple clarification can be done by blotting these native gels with antibodies to Syx, M18, Snb (as previously recommended). This would provide important clarification as to what the M18/Syx complex transitions to when MUN, Snb, SN25 are added. These data should be provided in the main text rather than as supplemental because of its central importance to the paper.

2. Text lines 227-230 on Fig 6C. The text in these lines does not seem to match what is shown in Fig 6C: "seemingly non-specific interaction between Syb2 and the M18/Syx complex that was not sensitive to the F77A mutation."

November 20, 2018

RE: NCOMMS-18-20580-A

Dear Reviewers,

Thank you very much for reviewing our manuscript. We are now submitting a revised manuscript where we have tried to address the concerns that the Reviewer #2 raised. We highlighted all changes with blue color in the revised main text.

Reviewer #2 (Remarks to the Author):

This manuscript has been improved by revision but there remains an important major concern (1) as well as a new minor concern (2).

1. Suppl Fig 1. The first part of this manuscript relies extensively on the assay shown in Fig. 1b that indicates a dissociation of a M18/Syx complex by the MUN domain. The dissociation required addition of Snb and SN25, and the authors term this dissociation a “transition” to the formation of Syx-containing SNARE complexes. However, there are other interpretations of these data. It was pointed out that SNARE complex formation was not evident in this native PAGE system. The authors responded by including Supp Fig 1 that annotates various protein bands in these reactions. Unfortunately these data are unclear because of the appearance of smeared bands. Of concern is the fact that the heavily-stained M18/Syx bands disappear and do not appear elsewhere and there is no corresponding increase that could represent the SNARE complexes. A very simple clarification can be done by blotting these native gels with antibodies to Syx, M18, Snb (as previously recommended). This would provide important clarification as to what the M18/Syx complex transitions to when MUN, Snb, SN25 are added. These data should be provided in the main text rather than as supplemental because of its central importance to the paper.

We appreciate the reviewer's comments. We have attached an immunoblotting result for the standard native-PAGE samples (**Supplementary Fig. 1**) (see below), and shifted the previous **Supplementary Fig. 1** to **Figure 1c**. Unfortunately, we currently do not have an appropriate Syb2/VAMP2 antibody for immunoblotting of Syb2 (residues 29–96). But we believe that the immunoblotting of Munc18-1 and Syx1 on the standard native-PAGE will provide clear interpretation for our native-PAGE assay.

Supplementary Figure 1. Immunoblotting of the standard native-PAGE assay. Syx1 and Munc18-1 antibodies were applied as indicated, respectively. Syx1 bound to Munc18-1 exhibits a strong and clear band, while free Munc18-1 displays smear-type band. The SNARE complex bands are as indicated. After the Munc18-1/Syx1 complex transits to the SNARE complex, Munc18-1 runs slightly faster than the isolated one (lane 14 vs. lane 2, lower panel), which suggests that there might have multiple interactions among Munc18-1, MUN, and the SNARE complex. Asterisk indicates potential aggregation of Syx1. Lane numbers are displayed at the top of the chart. In all conditions, Syx1 (residues 1–261) was 2 μ M, Munc18 was 2 μ M, SN25 was 10 μ M, Syb2 (residues 29–96) was 10 μ M, and MUN was 30 μ M.

2. Text lines 227-230 on Fig 6C. The text in these lines does not seem to match what is shown in Fig 6C: “seemingly non-specific interaction between Syb2 and the M18/Syx complex that was not sensitive to the F77A mutation.”

We feel sorry to make it unclear. Now the sentence has been fixed: “Our GST pull-down assay combined with immunoblotting detected a very weak interaction between Syb2 and the Munc18-1/Syx1 complex, and this interaction was not affected by the F77A mutation (Fig. 6c).” (Page 12, lines 226–228). -----

Sincerely,

Cong Ma
Huazhong University of Science and Technology

REVIEWERS' COMMENTS:

Reviewer #2 (Remarks to the Author):

The authors provide a set of studies that implicate MUN in catalyzing a transition of M18/Syx to a SNARE complex (Syx/Syb/SN25). Most of the work utilizes an assay in which M18/Syx disappears (when incubated with Syb, SN25, and MUN) shown in Fig 1b. In response to review, the authors provided current Fig 1c, a native gel analysis of the incubation products. As this Coomassie gel was ambiguous, the authors were requested to provide immunoblot data for MUN, Syb, Syx, and M18.

In response, the authors provided immunoblotting of native gels for Syx and M18 but not Syb or SN25. The results, shown in Supp Fig 1, are consistent with the authors' conclusion that SNARE complexes might form during the incubation although they remain ambiguous about whether Syb and SN25 are present in such complexes. Importantly, the results of Supp Fig 1 now somewhat compromise Fig 1c as it is currently labeled with a bracket called "SNARE complex" that covers multiple bands. It seems clear from Supp Fig 1 that Syx would correspond to only 1 or 2 of the bands. Other bands in Fig 1c labeled "SNARE complex" apparently lack Syx and would not be considered a traditional SNARE complex. Some comment by the authors or re-labeling of Fig 1c is needed to reconcile Fig 1c with Supp Fig 1

December 11, 2018

RE: NCOMMS-18-20580-B

Dear Reviewers,

Thank you very much for reviewing our manuscript. We are now submitting a revised manuscript where we have tried to address the concerns that the Reviewer #2 raised.

Reviewer #2 (Remarks to the Author):

The authors provide a set of studies that implicate MUN in catalyzing a transition of M18/Syx to a SNARE complex (Syx/Syb/SN25). Most of the work utilizes an assay in which M18/Syx disappears (when incubated with Syb, SN25, and MUN) shown in Fig 1b. In response to review, the authors provided current Fig 1c, a native gel analysis of the incubation products. As this Coomassie gel was ambiguous, the authors were requested to provide immunoblot data for MUN, Syb, Syx, and M18.

In response, the authors provided immunoblotting of native gels for Syx and M18 but not Syb or SN25. The results, shown in Supp Fig 1, are consistent with the authors' conclusion that SNARE complexes might form during the incubation although they remain ambiguous about whether Syb and SN25 are present in such complexes. Importantly, the results of Supp Fig 1 now somewhat compromise Fig 1c as it is currently labeled with a bracket called "SNARE complex" that covers multiple bands. It seems clear from Supp Fig 1 that Syx would correspond to only 1 or 2 of the bands. Other bands in Fig 1c labeled "SNARE complex" apparently lack Syx and would not be considered a traditional SNARE complex. Some comment by the authors or re - labeling of Fig 1c is needed to reconcile Fig 1c with Supp Fig 1.

We thank the reviewer's comments. We have re-labeled Fig.1c as suggested, and added the responding comments in the figure legend (see below) and in the main text (page 5, lines 86-90).

Figure 1. Half-zipped SNARE assembly gates MUN-catalyzed transition to the SNARE complex

a Illustration of the crystal structure of the SNARE complex (PDB entry: 3HD7). The lower panel displays the sequence of the SNARE motifs of Syx1, SN25, and Syb2. Hydrophobic binding layers -7 to +8 are indicated by rainbow-colored sticks in the upper panel and shaded in grey in the lower panel.

b Schematic diagram of the native-PAGE assay for monitoring MUN-catalyzed transition from the Munc18-1/Syx1 complex to the SNARE complex. The Munc18-1/Syx1 complex (2 μ M) displays a sharp band at the top of the gel; upon the addition of the MUN domain (30 μ M), SN25 (10 μ M) and Syb2 (10 μ M), this band disappears with the formation of the SNARE complex.

c Standard examples of the native-PAGE assay. The Munc13-1 MUN domain, Munc18-1 and free Syb2 show smeared band; free Syx1 displays multi-bands that likely represent different assembly/aggregation states; SN25 shows a strong and clear band; Syx1 bound to Munc18-1 exhibits a strong and clear band. Asterisk indicates a putative non-productive aggregation of the Munc18-1/Syx1 complex. **Octothorp indicates putative SNARE assembly intermediates that coexist with the actual ternary SNARE complex.** In the presence of Munc18-1 and the MUN domain, the SNARE complex displays a smear-type band that runs slower than the Munc18/MUN-free SNARE complex (lane 14 vs. lane 6). The intensity of the M18-1/Syx1 complex band does not change at all if lacking any of the component (lanes 8 –13); disappearance of the M18-1/Syx1 complex can only be detected when all the components are included (lane 14), indicating that the disappearance of the Munc18-1/Syx1 complex is accompanied by the appearance of the SNARE complex (a smear-type). Lane numbers are indicated at the top of the

chart. **d** Effects of Syb2 N-terminal mutations or truncations on MUN-catalyzed transition from the Munc18-1/Syx1 complex to the SNARE complex using native PAGE. The representative gel displayed is from one of three replicates. **e** Quantification of (**d**). Integrated density represents the normalized integrated grey level of each assessed Munc18-1/Syx1 band. Data are presented as the means \pm SD, n = 3, two-tailed t-test, ***, p<0.001. **f** Effects of SN25 C-terminal truncations on MUN-catalyzed transition from the Munc18-1/Syx1 complex to the SNARE complex using native PAGE. The representative gel displayed is from one of three replicates. **g** Quantification of (**f**). Integrated density represents the normalized integrated grey level of each assessed Munc18 - 1/Syx1 band. Data are presented as the means \pm SD, n = 3, two-tailed t-test, **, p<0.01; ***, p<0.001. Source data are provided as a Source Data file.

Sincerely,
Cong Ma
Huazhong University of Science and Technology